



# Above-aircraft cirrus cloud and aerosol optical depth from hyperspectral irradiances measured by a total-diffuse radiometer

Matthew S. Norgren[1], John Wood[2], K. Sebastian Schmidt[1,3], Bastiaan van Diedenhoven[4,5], Snorre A. Stamnes[6], Luke D. Ziemba[6], Ewan C. Crosbie[6], Michael A. Shook[6], A. Scott Kittelman[3], Samuel E. LeBlanc[7,8], Stephen Broccardo[7,8,9], Steffen Freitag[10], Jeffery S. Reid[11]

[1]Laboratory for Atmospheric and Space Physics, University of Colorado, Boulder, CO, USA
[2] Peak Design Ltd, Sunnybank House, Wensley Rd, Winster, Derbys, DE4 2DH, UK
[3] Department of Atmospheric and Oceanic Sciences, University of Colorado, Boulder, USA
[4] SRON Netherlands Institute for Space Research, Utrecht, Netherlands
10   [5] NASA Goddard Institute for Space Studies, New York, NY, USA
[6] NASA Langley Research Center, Hampton, VA, USA
[7] Bay Area Environmental Research Institute, Moffett Field, CA, USA
[8] NASA Ames Research Center, Moffett Field, CA, USA
[9] Universities Space Research Association, Columbia, Maryland, USA
15   [10] Department of Oceanography, University of Hawaii at Manoa, Honolulu, HI, USA
[11] Marine Meteorology Division, U.S. Naval Research Laboratory, Monterey, CA, USA

*Correspondence to*: Matthew S. Norgren (matthew.norgren@colorado.edu)

**Abstract.** This study develops the use of spectral total and diffuse irradiance measurements, made from a prototype hyperspectral total-diffuse Sunshine Pyranometer (SPN-S), to retrieve layer fine-mode aerosol ($\tau_{aer}$) and total optical depths from airborne platforms. Additionally, we use spectral analysis in an attempt to partition the total optical depth it into its $\tau_{aer}$ and cirrus cloud optical depth ($\tau_{cld}$) components in the absence of coarse-mode aerosols. Two retrieval methods are developed: one leveraging information in the diffuse irradiance, and the other using spectral characteristics of the transmitted direct beam, with each approach best suited for specific cloud and aerosol conditions. SPN-S has advantages over traditional sun-photometer systems including no moving parts and a low cost. However, a significant drawback of the instrument is that it is unable to measure the direct beam irradiance as accurately as sun-photometers. To compensate for the greater measurement uncertainty of the radiometric irradiances these retrieval techniques employ ratioed inputs or spectral information to reduce output uncertainty. This analysis uses irradiance measurements from SPN-S and the Solar Spectral Flux Radiometer (SSFR) aboard the National Aeronautics and Space Administration's (NASA) P-3 aircraft during the 2018 deployment of the ObseRvations of Aerosols above CLouds and their intEractionS (ORACLES) campaign and the 2019 Cloud, Aerosol and Monsoon Processes Philippines Experiment (CAMP²Ex) mission to quantify above-aircraft cirrus $\tau_{cld}$ and derive vertical profiles of layer $\tau_{aer}$. Validation of the $\tau_{aer}$ retrieval is accomplished by comparison with collocated measurements of direct solar irradiance made by the Sky-Scanning Sun-Tracking Atmospheric Research (4STAR) and in situ measurements of aerosol optical depth. For the aggregated 2018 ORACLES results, regression between the SPN-S based method and sun-photometer $\tau_{aer}$ values yield a slope of 0.96 with an $R^2$ of 0.96, while the root-mean-square error (RMSE) is $3.0 \times 10^{-2}$. When comparing





the retrieved $\tau_{aer}$ to profiles of integrated in situ measurements of optical extinction, the slope, $R^2$, and RMSE values for ORACLES are 0.90, 0.96, $3.4 \times 10^{-2}$, and for CAMP²Ex are 0.94, 0.97, $3.4 \times 10^{-2}$ respectively.

This paper is a demonstration of methods for deriving cloud and aerosol optical properties in environments where both atmospheric constituents may be present. With improvements to the low-cost SPN-S radiometer instrument, it may be possible to extend these methods to a broader set of sampling applications, such as ground-based settings.

## 1 Introduction

     Clouds and aerosol particles both play important roles in controlling the flux of solar radiation through the Earth's
atmosphere. Despite their relevance to the broader climate system and Earth's radiative balance, significant uncertainty exists in quantifying the optical properties of atmospheric systems containing one or both constituents. Traditional passive remote sensing methods retrieve aerosol properties in the absence of clouds (Holben et al., 1998; Levey et al., 2013). This is because when clouds are thick their radiative signal is large in relation to the aerosol signal, and when clouds are thin it is difficult to separate the two signals. For automated aerosol optical depth ($\tau_{aer}$) retrievals, the challenge of cloud detection and removal is
a significant hurdle to overcome (Smirnov et al., 2000; Remer et al., 2012; Spencer et al., 2019). Advanced methods, such as the spectral deconvolution algorithm (SDA) have been developed to differentiate between fine- and coarse-mode $\tau_{aer}$ using spectral sun-photometry data (O'Neill et al., 2003), though these techniques are limited when cirrus are present (Smirnov et al., 2018). Conversely, retrieval of cloud optical depth ($\tau_{cld}$) tends to be insensitive to the aerosol loading of the local environment because clouds are often much thicker optically than aerosols. A common exception to this occurs when thin
cirrus clouds are present. Reported values for mean cirrus cloud optical depth vary regionally (Giannakaki et al., 2007; Dai et al., 2019), but tend to be less than a value of unity (Kox et al., 2014, Heymsfield et al., 2017; Zhou et al., 2018). Since it is common for cirrus to have optical depths similar to those of aerosols, which typically have values less than 4 in equatorial regions (Torres et al., 2002), remote sensing of the optical depth of either constituent is complicated by the presence of the other.

The necessity of retrieving both aerosol and cirrus cloud optical properties is supported by the fact that aerosols are ubiquitous throughout Earth's atmosphere and cirrus clouds are globally prevalent (Sassen et al., 2008). Cirrus presence is especially high in equatorial regions where their frequency of occurrence can be near 50 percent. Overlying cirrus limits remote sensing of cloud and aerosol properties by passive airborne radiometers and polarimeters (e.g., Werner et al. 2013; Stamnes et al. 2018), and ground-based sun-photometer retrievals of $\tau_{aer}$ frequently suffer from contamination by these clouds
(Chew et al., 2011). In the past, attempts have been made to account for and correct retrievals of $\tau_{aer}$ for the impact of cirrus (Lee et al., 2013). Efforts have also been made to use sun-photometry to derive cirrus $\tau_{cld}$ (Kinne et al., 1997; Segal-Rosenheimer et al., 2013), but work towards joint $\tau_{cld}$ and $\tau_{aer}$ retrieval is limited.

     In this paper we address some of the issues associated with remote sensing of thin cloud and aerosol systems by leveraging the capabilities of a new hyperspectral total-diffuse radiometer, SPN-S. The advantage of this radiometer system is



that it is low-cost and deployable to a wide range of environments. The device is mechanically simple, with a fixed shadow mask used to block the direct beam of the Sun to make measurements of the diffuse irradiance. This shadow mask design allows for simultaneous measurements of both the total and diffuse fluxes, which is functionality that traditional rotating shadow band radiometer systems cannot obtain. The concurrent sampling of the two irradiances is useful in airborne, or other dynamic, settings where scenes can change rapidly. However, radiometer systems like SPN-S, have higher measurement

uncertainties than sun-photometer systems, which limits their ability to investigate atmospheric optical properties. To compensate for lower accuracy, we propose a method for deriving $\tau_{cld}$ of thin clouds using narrowband measurements of the diffuse to total ratio ($DR$). This Diffuse Ratio Method, which will be referred to as RD, is advantageous for the study of thin clouds ($\tau < 1$) because its main radiometric input is a ratio of two measured irradiances made from the same instrument, which reduces absolute calibration-induced errors. This use of the ratio leaves the main sources of uncertainty as the instrument

precision and assumptions made in the retrieval itself, resulting in RD being highly sensitive to small variations in optical depth. A second method, that we refer to as the Spectral Direct Beam Method (RS), uses measurements of the direct irradiance to develop optical depth spectra. The shape of the spectral optical depth curve contains information on the loading of fine-mode and large, coarse-mode particles in the atmosphere. In the absence of coarse-mode aerosols we show the potential to retrieve $\tau_{aer}$ and $\tau_{cld}$ values.

The two methods presented in this study overlap with previous cloud and aerosol retrieval techniques. DR has been used to study aerosol single-scattering albedo and asymmetry parameter properties using measurements from Multi-filter Rotating Shadow Band Radiometers (MFRSR; Kassianov et al., 2007; Herman et al., 1975). Although MFRSR is the most widely-used total-diffuse radiometer system, past work utilizing the instrument has relied on the direct irradiance measurement when deriving $\tau_{aer}$ (Michalsky et al., 2010). Likewise, the use of spectral shape of the transmitted direct beam in the RS

method is similar to the SDA method developed by O'Neill et al. (2003). However, MFRSR and sun-photometer systems require extensive alignment and precise operating conditions, making these past methods not applicable to airborne settings and restricting them to use at the surface. Extending these previous works, and then using them in conjunction with a shadowmask-designed radiometer, allows for a broader set of applications. Specifically, the use of the SPN-S in airborne settings (or other non-stationary environments) is a novel application of a spectral total-diffuse radiometer system which allows

for greater detail of the atmospheric aerosol and cloud structure to be known.

        This paper describes the theoretical underpinnings of the retrievals in Section 2, along with a justification of which method is best suited for specific atmospheric conditions. Section 3 overviews the data used in the study from two airborne field campaigns: the 2018 deployment of the ObseRvations of Aerosols above CLouds and their intEractionS (ORACLES) campaign conducted over the Southeast Atlantic Ocean, and the 2019 Cloud, Aerosol and Monsoon Processes Philippines

Experiment (CAMP²Ex) mission conducted above the waters surrounding the Philippines. Section 3 follows with a detailed accounting of the retrieval algorithm implementation using the field measurements. Section 4 presents the results: first a comparison of retrieved $\tau_{aer}$ values to co-located $\tau_{aer}$ measurements made by a sun-photometer as well as in situ measurements of the aerosol optical depth. Then summary statistics of the ORACLES and CAMP²Ex campaigns are shown.





Section 5 is a discussion of limitations and usefulness of the two new methods. Section 6 provides a brief summary of the
manuscript.

## 2 Theory and approach

The attenuated direct solar radiation in a layered model of the atmosphere is directly related to the layer optical depth
by Beer's Law:

$$F_{dir} = F_0 e^{-\tau/\mu} \qquad \text{Eq. 1}$$

where $F_{dir}$ is the direct beam irradiance, $F_0$ is the incident irradiance at the top of the layer, $\mu$ is the airmass factor which we
approximate as the cosine of the solar zenith angle (SZA), and $\tau$ is the layer optical depth. Inverting Equation 1 is frequently
invoked to derive optical depth from measurements of $F_{dir}$.

Conversely, the two-stream approximation of the Radiative Transfer Equation can be used to relate the downwelling
diffuse irradiance to the optical depth (Bohren and Clothiaux, 2006, eq.5.66, p.263):

$$\frac{F_{dif}}{F_0} = \frac{1}{1 + \tau(1-g)/2\mu} - e^{-\tau/\mu} \qquad \text{Eq. 2}$$

where $g$ is the asymmetry parameter of the single atmospheric layer (a parameter used in the two-stream approximation to
describe the relative amounts of forward and backwards scattering within a layer), and $F_{dif}$ is the downwelling diffuse
irradiance. It is important to note that Eq. 2 assumes no absorption and the atmospheric layer is above a black surface. By
solving Eq. 1 for $F_0$, substituting this into Eq. 2, and consolidating all the irradiance terms to the left-hand side, we get an
expression relating the diffuse ratio (DR) to optical depth:

$$Diffuse\ Ratio\ (DR) = \frac{F_{dif}}{F_{dir} + F_{dif}} = 1 - (1 + \tau(1-g)/2\mu)e^{-\tau/\mu} \qquad \text{Eq. 3}$$

In the thin cloud limit:

$$\frac{\tau(1-g)}{2\mu} \ll 1 \qquad \text{Eq. 4}$$

Eq. 3 simplifies to:

$$DR = 1 - e^{-\tau/\mu} \qquad \text{Eq. 5}$$

and the dependence of $DR$ on $g$ is minimal. Crucially, Eq. 3 and Eq. 5 are not dependent on $F_0$, which allows for knowledge
of $\tau$ to be obtained without information (or assumptions) of the irradiance incident upon the layer. These equations give us two
relationships between observables, $DR$ and $F_{dir}$, to optical depth. Eq. 1 and Eq. 3 form the basis of the two retrieval methods:
(1) using $DR$ to derive $\tau_{cld}$; and (2) exploiting spectral features of $F_{dir}$ to partition $\tau$ into $\tau_{cld}$ and $\tau_{aer}$ components.

### 2.1 Diffuse Method, RD − $\tau_{cld}$ retrieval

Equation 3 directly links $DR$ to optical depth for cases when the extinction of the layer is solely caused by scattering.
Cloud particles have minimal absorption coefficients for light at visible wavelengths (Bohren and Huffman, 1998) and
therefore a single-scattering albedo (SSA) near unity. Fine-mode aerosols are commonly absorbing, while coarse-mode



aerosols, such as sea-salt and dust, have similar absorption characteristics to clouds, both of which limits the application of
RD to samples without aerosols (the implications aerosols have on RD are discussed in more detail in Section 5). Given these
constraints, Eq. 3 can be solved to derive cloud optical depth. We apply this model specifically to retrieve cirrus cloud optical
depth because, as we will show in Section 2.3, $DR$ is most sensitive to changes in $\tau$ when $\tau$ is small. At optical depths $\tau_{cld} >$
$\sim5$, $DR$ asymptotes to unity, leaving little information on $\tau$.

At smaller cloud optical depths, when direct beam is still present ($\tau_{cld} < \sim5$), $DR$ is linked to the amount of forward
scattering promoted by the cloud medium, and therefore a source of uncertainty in RD is incomplete knowledge of $g$ (or the
full scattering phase function). For cirrus clouds $g$ typically ranges from 0.7 to 0.9 (Fu, 2007). Likewise, surface albedo, $a$,
impacts $DR$ through a process of multiple scattering of light between the surface and the cloud layer. Since the analytic solution
in Eq. 3 does not account for the impact of surface albedo on $DR$, we use a radiative transfer model (RTM) to accurately
represent the sampled environment and implement the retrieval. In the field settings, the flight-level $a$ ($a_{fl}$) is directly inferred
by ratioing downwelling and upwelling total irradiance measured by SPN-S and Solar Spectral Flux Radiometer (SSFR)
respectively.

The retrieval steps or the RD method are simple:
1. Given a measurement of diffuse ratio, $DR_{mea}$, we solve Eq. 5 to make an initial estimate of the cloud optical depth,
$\tau_{est}$.
2. $\tau_{est}$ and measured flight-level albedo, $a_{fl}$, are used as inputs into the RTM to simulate a diffuse ratio, $DR_{sim}$. $g$ is set
to 0.85 in the RTM.
3. $DR_{sim}$ to $DR_{mea}$ are compared:
    3.1. If the difference is greater than $\pm1\%$ $\tau_{est}$ is adjusted by a small amount, $\tau_{est} = \tau_{est} \pm \Delta\tau$, and step 2 is repeated.
3.2. If the difference is less than $\pm1\%$, $\tau_{cld} = \tau_{est}$.
4. The $\tau_{cld}$ value has bias induced by the wide field-of-view (FOV) of the SPN-S, and this bias is corrected for in certain
sampling settings. Details of the FOV correction are found in Section 3.2 and Appendix A.

The details and specifics of the measurements and RTM used in the diffuse method, RD, will be discussed in Section 3 of this
paper.

**2.2 Spectral Direct Beam Method, $RS - \tau_{cld}, \tau_{aer}$ retrieval**

In The Spectral Direct Beam Method, RS, leverages the differences in the spectral dependence of layer optical depth
containing small (fine-mode) and large (coarse-mode) particles. Layers containing small particles, with sizes roughly that of
the wavelength of visible light, have a strong wavelength dependence in optical depth. For fine-mode aerosols $\tau_{aer,\lambda}$ can be
described by:

$\tau_{aer,\lambda} = \tau_{aer,0} \left(\frac{\lambda}{\lambda_0}\right)^{-AE}$ 
                                                                  Eq. 6





where the lambda subscript of the $\tau_{aer,\lambda}$ term indicates a spectral dependence to the optical depth, $\tau_{aer,0}$ is the optical depth at a reference wavelength $\lambda_0$, and $AE$ is the Ångström exponent (Ångström, 1929).

Larger particles, such as coarse-mode aerosol (e.g., sea salt or dust) and cloud hydrometeors, reside close to, or in the geometric scattering regime at visible wavelengths and therefore have low $AE$ value. This results in minimal wavelength dependence of $\tau$ (i.e., are largely spectrally flat in the visible) of atmospheric layers containing these large particles. In an atmospheric system containing fine-, but no coarse-, mode aerosols, we expect the spectral total layer optical depth to be of the form:

$$\tau_\lambda = \tau_{cld,\lambda} + \tau_{aer,\lambda} = \tau_{cld} + \left[\tau_{aer,0}\left(\frac{\lambda}{\lambda_0}\right)^{-AE}\right] \qquad \text{Eq. 7}$$

In practice we use Beer's Law, Eq. 1, to determine $\tau_\lambda$, where this spectral optical depth is also represented by the summation of all atmospheric constituents with non-zero optical depth:

$$\tau_\lambda = \tau_{cld,\lambda} + \tau_{aer,\lambda} + \tau_{Ray,\lambda} + \tau_{mol,\lambda} \qquad \text{Eq. 8}$$

where $\tau_{Ray,\lambda}$ is the spectral optical depth from Rayleigh scattering, and $\tau_{mol,\lambda}$ is a term encompassing non-Rayleigh extinction from trace gas molecular scattering and water vapor absorption sources. In the wavelength range used in this analysis there is optical depth from trace gases and water vapor, but rather than measure or calculate the $\tau_{mol,\lambda}$ term, we select wavelengths that minimize its value. Further, we apply a correction to the derived $\tau_\lambda$ based on measured values of $\tau_\lambda$ from above the aerosol layer (see Section 3.5.2). That is, we are calculating a layer aerosol optical depth, and in doing so we account for much of the influence of $\tau_{mol,\lambda}$ on $\tau_\lambda$. Future work may want to treat the $\tau_{mol,\lambda}$ term with more detail which would possibly allow for column, and not layer, $\tau_{aer,\lambda}$ to be retrieved. Then if wavelengths are selected that minimize $\tau_{mol,\lambda}$, $\tau_\lambda$ is dependent on $\tau_{cld}$, $\tau_{aer,\lambda}$, and $\tau_{Ray,\lambda}$. $\tau_{Ray,\lambda}$ can be solved for empirically and is a function of the pressure differential across the observed layer:

$$\tau_{Ray,\lambda} = c_1\left[\frac{c_2 - c_3\lambda^2 - c_4\lambda^{-2}}{1 - c_5\lambda^2 - c_6\lambda^{-2}}\right]\left(\frac{\Delta P}{1013.25}\right) \qquad \text{Eq. 9}$$

where $\Delta P$ is the pressure differential across the layer in millibar, $c_1 = 2.10966 \times 10^{-3}$, $c_2 = 1.0455996$, $c_3 = 341.29061$ $\mu$m$^{-2}$, $c_4 = 9.02308508 \times 10^{-1}$ $\mu$m$^{-2}$, $c_5 = 2.7059889 \times 10^{-3}$ $\mu$m$^{-2}$, $c_6 = 85.968563$ $\mu$m$^{-2}$ (Hansen and Travis 1974). Using Eq. 9 to calculate and account for the $\tau_{Ray,\lambda}$ term in Eq. 8, the spectral shape of the layer optical depth dictated by Eq. 7 can be described in terms of $\tau_{cld}$ and $\tau_{aer,\lambda}$. An example of this is illustrated in Figure 1 for simulated optical depths (RTM configuration is given in Section 3.3) for a case with a cloud only (blue lines) and cloud with aerosol (red lines). A set of selected retrieval wavelengths, designated by the yellow shading, represents regions of the spectrum where $\tau_{mol}$ is minimal (additionally the shaded regions correspond with usable channels from SPN-S; more in Section 3). For the aerosol-free case within the selected wavelengths region, the $\tau_\lambda$ profile (dashed blue line) falls nearly along a flat line of the simulated cloud optical depth value of $\tau_{cld} = 0.20$. When fine-mode aerosols are present, $\tau_\lambda$ is curved in the form of Eq. 7, and asymptotes to a value of $\tau_{cld} = 0.20$ at longer wavelengths. The aerosol optical depth is simply the difference in the layer and cloud optical depths at 500 nm: $\tau_{aer,500nm} = \tau_{500nm} - \tau_{cld}$.





In practice RS is implemented as follows:

1. Spectral measurements of direct irradiance, $F_{dir}$, are used to determine $\tau_{mea,\lambda}$ using Beer's Law, Eq. 1.


2. Wavelengths in window channels are selected that minimize the $\tau_{mol,\lambda}$ term in Eq. 8.

3. $\tau_{mea,\lambda}$ is characterized above the aerosol layer and used to correct all profile samples of $\tau_{mea,\lambda}$ for the influence of $\tau_{mol,\lambda}$.

4. $\tau_{Ray,\lambda}$ is calculated from Eq. 9 and then subtracted from the corrected $\tau_{mea,\lambda}$.

5. A set of calculated layer optical thicknesses, $\tau_{calc,\lambda}$, are found using Eq. 7 for a range of $\tau_{cld}$, $\tau_{aer,0}$, and $AE$

values.

6. The root-mean-square error (RMSE) is found for each combination of $\tau_{calc,\lambda}$ and $\tau_{mea,\lambda}$ profiles. The retrieval outputs— $\tau_{cld}$, $\tau_{aer}$, and $AE$ —are the values corresponding to the $\tau_{calc,\lambda}$ with the lowest RMSE.

7. Field-of-view correction applied to the $\tau_{cld}$ retrieval output (see Section 3.2 and Appendix A).

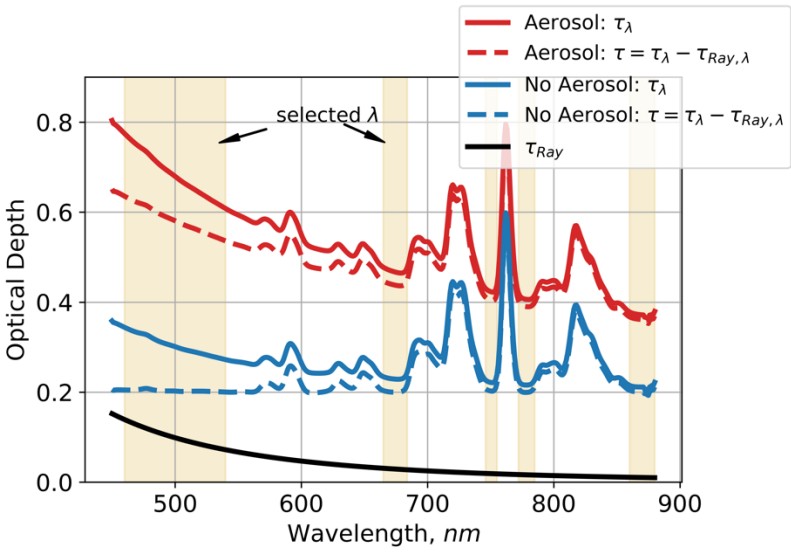

**Figure 1:** Simulated spectral $\tau_\lambda$ for two cases: cloud only, $\tau_{cld} = 0.20$, $\tau_{aer,500nm} = 0.00$, is denoted by the blue lines. And a case with cloud and aerosols, $\tau_{cld} = 0.20$, $\tau_{aer,500nm} = 0.38$, shown by the red lines. The solid black line is calculated $\tau_{Ray,\lambda}$. Subtracting $\tau_{Ray,\lambda}$ from $\tau_\lambda$ yields the dashed red and blue lines, which are represented by Eq. 7. The shaded yellow regions are the wavelengths selected for use in RS. For the aerosol free case, the layer optical depth is spectrally flat at the value of $\tau_{cld}$.



**2.3 RD and RS comparison and use selection consideration**

Without knowledge of the output uncertainties of the two methods, the separation of the aerosol from the cloud radiative signal makes RS the more capable of the two retrieval methods. However, RS is based on measuring $F_{dir}$, which is a measurement that is susceptible to significant errors when made from airborne platforms. For SPN-S, the main error sources of the irradiance measurements are the result of improper radiometric calibration, the sensor cosine response, and errors associated with the attitude of the sensor relative to the horizon. RD *has the advantage of being derived from the ratio of two*

*irradiances simultaneously measured by the same instrument.* In the case of SPN-S, the diffuse and total irradiance measurements are made by the same sensor which allows us to assume the radiometric uncertainties of $F_{dif}$ and $F_{tot}$ are correlated, and therefore the uncertainty of $DR_{mea}$ is a function of the instrument precision, and not the accuracy. Further, DR is minimally affected by sensor attitude errors because the $F_{dir}$ term is in the denominator. For SPN-S irradiance measurements we estimate a $4 - 6\%$ accuracy uncertainty results from the lamp calibration process and up to another 2% uncertainty stems

from imperfect knowledge of the cosine response of the sensor. For the sake of this analysis, we will use uncertainty values of 7% accuracy and 0.5% for precision (Wood et al., 2017). There are additional sources of error (e.g., changes in sensor attitude) that will be addressed in the following sections.

The performance of both retrieval methods is gauged considering the uncertainty of the inputs. First, we evaluate Eqns. 1 and 3 for a range of $\tau$ values. Figure 2 shows the profile of the direct transmittance, $T_{direct} = F_{dir}/F_0$ (green line),

given a 7.0% uncertainty in $F_{dir}$. At high values of $T_{direct}$ (low $\tau$), the measurement uncertainty is a significant proportion of the signal, leading to substantial ambiguity in the associated value of $\tau$. As optical depths increase, $T_{direct}$ falls off exponentially at the same rate as its error, meaning the absolute error in $\tau$ is constant, and hence the fractional error decreases as the cloud layer thickens. For RD the uncertainty on $DR$ is minimal, and so for a known value of $g$ the retrieved $\tau$ has little ambiguity until the $DR$ signal starts to asymptote to unity at values $\tau > \sim 5$. However, the uncertainty in the value of $g$ is a

main source of error when determining $\tau$ at low optical depths. The pink and purple lines are $DR$ profiles for $g$ ranging from 0.5-1.0. At low values of $DR$, when scattering is minimal, the uncertainty in $g$ leads to relatively little ambiguity in the value of $\tau$ (i.e., $g$ is not represented in Eq. 5). As the cloud optical depth increases, greater amounts of scattering occur and the importance of $g$ on $DR$ becomes more pronounced, leading to poorer retrieval performance. Fortunately, the two methods for deriving $\tau$ are complimentary – RD has lower uncertainty in retrieving thin cloud $\tau$ whereas the use of RS is justified as $\tau$

becomes larger. This threshold at which the output uncertainty of RS falls below RD is approximately when $\tau \sim 1$, though this level is dependent on how well $g$ is constrained and the level of uncertainty of the inputs. Both methods lose utility as $\tau$ becomes large and the light becomes completely diffuse.

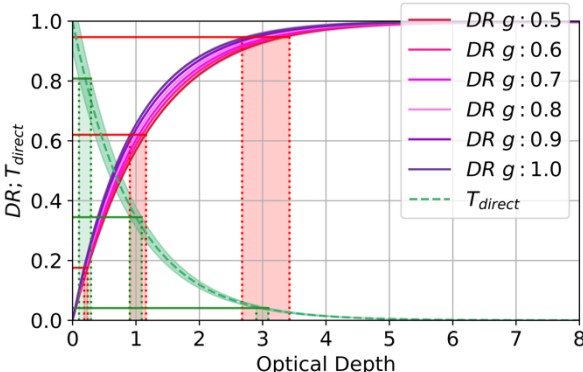

**Figure 2:** Solutions given the direct transmittance using Beer's Law (Eq. 1) and the diffuse ratio (Eq. 3) giving layer optical depth. Shown are uncertainties in optical depth for irradiances associated with layer optical depths of 0.2, 1.0, and 3.0. Uncertainty in $F_{dir}$ is 7.0%, and uncertainty in $DR$ is 0.5%.

To further address the question of retrieval performance, we test the explicit retrieval methods used in this study (outlined in Sections 3.4) on model-generated irradiance data. To do this we simulated a set of spectral irradiances using a RTM (see Section 3.3) for values of $\tau_{cld}$ ranging from 0 to 6. The cloud optical properties in the simulations were generated using the "hey" ice cloud parameterizations that are a part of the public libRadtran package (Yang et al., 2013; Emde et al., 2016). For this example, a cloud comprised of smooth solid-column ice crystals with an effective radius of 20 $\mu$m was inserted

in the atmosphere with a base height of 10 km, and we set $a = 0.15$. In practice, we directly infer $a$ using SPN-S and SSFR measurements, so in this exercise we do not investigate the influence of albedo on the retrieval error. The $DR$ profiles are calculated at three values of $g$: 0.70, 0.85, 0.95. The simulated inputs, $F_{dir}$ for RS and $DR$ for RD, are then injected into both retrieval structures, and the outputs compared to the true values of $\tau_{cld}$ – the values of $\tau_{cld}$ set in the RTM simulations.

Figure 3 shows the results of this experiment. The left panel is the simulated 500 nm downwelling irradiances. The

middle panel shows the retrieved values of $\tau_{cld}$ plotted against the true $\tau_{cld}$, and the right panel is the error in retrieved $\tau_{cld}$ value as a function of true $\tau_{cld}$. The results support the analysis of the analytic functions presented in Figure 2: RD outputs have less error and RD is therefore the superior method when $\tau_{cld} < {\sim}1$, and RS retrieves $\tau_{cld}$ with less error for optical depths above unity. It is critical to note that this analysis of cloud simulations assumes an absence of aerosols. If absorbing aerosols are present, knowledge of SSA and $g$ are needed to accurately use RD to retrieve optical depth.

One other limitation that is worth mentioning: the output of RD fall below the identity line because of difference in the scattering phase functions used in the simulated data and the $DR$ calculations in the retrieval. In RD, the phase function is approximated using the Henyey-Greenstein phase function for a given value of $g$, while the simulated data uses full phase functions for clouds layers containing solid-column ice crystal habit from the libRadtran library. We use Henyey-Greenstein





in the retrieval to make it more broadly applicable to various atmospheric conditions and to avoid making assumptions about

the cloud microphysics.

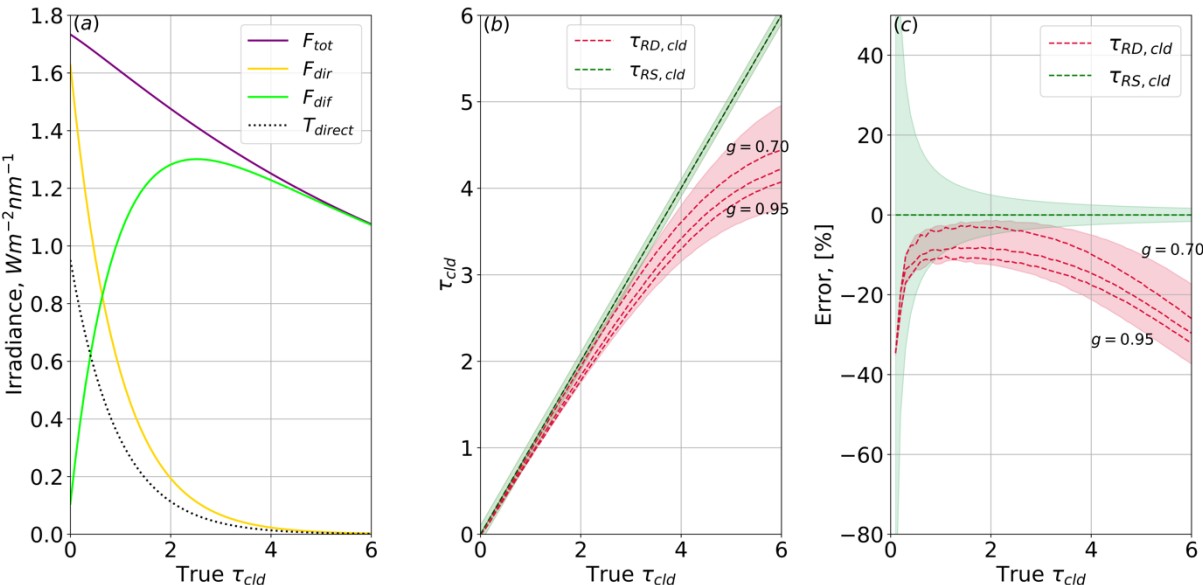

**Figure 3:** (a) Simulated 500 nm irradiances at 4 km altitude, with an ice cloud situated at 10 km. Solar zenith angle is 20 degrees, ice crystal effective radius is 20 $\mu$m, and the model output level is 4 km. (b) Retrieval outputs with the simulated irradiances used as inputs. Error of $F_{dir}$ is 7.0%, error of $DR$ is 0.5%. The uncertainty in retrieval output is represented by the shading. For RD, this uncertainty also reflects $g$ values of 0.70, 0.85, 0.95. (c) Percent error of the two retrieval methods when compared to the true cloud optical depth.

## 3 Data and implementation:

This section details the data sources used in the retrievals and in validation of their outputs. Additionally, the RTM
used throughout this study is described. This is followed by a detailed explanation of the procedures used to implement RS and RD with the field data.

### 3.1 ORACLES and CAMP²Ex campaigns

The data used in this study come from two independent field campaigns that occurred in climatically different tropical
to subtropical regions of the globe. Both campaigns used the National Aeronautics and Space Administration's (NASA) P-3 aircraft equipped with a set of radiometric and in situ scientific instruments, with the goal of addressing questions surrounding cloud and aerosol impacts on atmospheric state. Specifically, ORACLES was a three-part mission focused on studying the radiative effects of biomass burning generated aerosols present above clouds. Sampling of the aerosol plume was done over



the eastern Atlantic Ocean after it had advected off the west coast of the African continent. The experiment environment at ORACLES was stratified – with a persistent aerosol layer sitting below clear-skies (i.e., minimal cirrus clouds). A stratus cloud layer, of varying cloud fraction, was often found below or at the bottom of the aerosol layer. While cirrus did tend to occur in the sampling region, flight planning was done to avoid their presence. This environment allowed for measurements of aerosol properties to be made with minimal influence from surrounding clouds, outside of the impact the low cloud layer had on albedo. The science flights took place in August-October of 2016-2018. The SPN-S was only developed in time for operation

during the 2018 deployment of the ORACLES mission. A complete overview of the ORACLES campaign is given by Redemann et al. (2020).

In contrast, CAMP²Ex took place from late August through early October of 2019, in the maritime environments surrounding the island of Luzon, Philippines. This location was selected because polluted airmasses from Borneo and Sumatra could be characterized as they are transported though the South China Sea and Sulu Sea into the Western Pacific by the

Maritime Continent's Southwest Monsoon flow. Sampling took place in several different airmasses which had variability in the aerosol source – local sources from industrial activity on Luzon (especially from Manila) and tanker ships, to biomass burning aerosols transported from Borneo, and long-range transport of aerosols from the Asian continent. Given the nature of the Southwest Monsoon, the cloud environment was highly dynamic, with cumulus clouds and convective cells of varying degrees of maturity present during all flight days. Unlike with ORACLES where cirrus could be avoided, during CAMP²Ex

regional deep convection led to ubiquitous cirrus. It is this high prevalence of the cirrus clouds during the mission which provided much of the motivation for developing the spectral approaches for determining $\tau_{cld}$ and $\tau_{aer}$. CAMP²Ex also sampled for 10 days after the monsoon transition, leading to lower cirrus optical depths and sampling of the heavily polluted Asian airmasses. For an overview of the CAMP²Ex mission, refer to Reid et al., (2021).

A note on sampling methods: during both campaigns we employed a "square-spiral" sampling technique to profile

layers of the atmosphere up to two times per science flight. This is where the P-3 aircraft descends through the atmosphere, from high altitude flight to near surface, by making a box pattern. The box pattern consisting of four legs of decent, with wings held level (i.e., pitch and roll kept as close to 0° as possible), connected by 90° degree descending turns that are banked. The goal is to minimize the influence aircraft attitude has on the radiometers position relative to the horizon, thereby reducing the magnitude of the attitude correction that is needed in post-processing of the radiometric data. In comparison to a series of

traditional flat "radiation legs", the "square-spiral" method allows for relatively rapid profiling of the atmosphere, which is useful in dynamic environments such as the ones encountered in CAMP²Ex. At CAMP²Ex near-surface clouds were encountered in most spirals, and so for this study we restricted spiral profiles to altitudes greater than 0.4 km. At ORACLES the stratus deck was mostly found below1 km, and we limit sampling to above this altitude.

**3.2 SPN-S irradiances and retrieval input data**

The SPN-S is a modified version of the commercially available broadband SPN1 pyranometer. Instead of the detector head directing the sampled light to a set of 7 thermopiles, this spectral version uses a 7-spectrometer array to measure irradiance





from 350-1000 nm at 1 nm spectral resolution and 1 Hz temporal resolution. Detailed characterization of SPN1, and how the measurements of total and diffuse irradiance are made, is described in Badosa et al. (2014). The version of the SPN-S used in

this study is most similar to the Spectrometer 1 system described in Wood et al. (2017), with modifications done to the instrument chassis to allow it to be mounted in the top of the P-3 fuselage. We follow the procedure laid out in Wood et al. (2017) for deriving the spectral direct irradiance from the measurements of total and diffuse irradiance.

Our procedure for calibration and attitude correction deviates from the methods described in Wood et al., (2017). The SPN-S was calibrated against a tungsten 'FEL' lamp that is traceable to a National Institute of Standards and Technology

(NIST) standard. The power of the 'FEL' lamp is significantly lower than the power of the solar radiation at the sampling sites and this led to issues with measuring irradiances at the shorter and longer wavelengths of SPN-S's capabilities. For our analysis we only use sampled radiation from 460-900 nm due to these calibration constraints.

The direct beam and total irradiances were corrected for the pitch and roll of the aircraft in accordance with the attitude correction method outlined in Long et al., (2010). Due to time limitations of the P-3 aircraft and cloud cover constraints,

we could not fly the recommended "box" and "diagonal" patterns needed to determine pitch and roll offset angles of the mounted SPN-S. Rather, we manually went through all the flight data from the ORACLES campaign, identified heading changes of the aircraft that occurred under clear-sky conditions ($DR < 0.1$), and used the collective set of these heading changes as a substitute for the "box" and "diagonal" patterns in the Long et al. (2010) method. Since SPN-S was mounted on the P-3 in the same configuration for the CAMP[2]Ex mission as it was during ORACLES, we assume the offset angles are

constant across the two missions. To minimize the impact aircraft attitude has on the direct irradiance measurements we restrict our analysis to when both aircraft pitch and roll were less than $\pm 3$ degrees.

To measure diffuse irradiance, the SPN-S uses a shadow mask to block the direct beam of the sun. The field-of-view (FOV) blocked by the sun is an area larger than the solar disk, and this wide FOV leads to understatement of the diffuse irradiance under sky conditions where there is a significant amount of light scattered in the direction of the direct beam

(conversely, direct transmittance is overstated). This is similar to FOV issues encountered by sun-photometer systems (Segal-Rosenheimer et al., 2013). Thin cirrus cloud layers are associated with strong forward scattering, and the induced bias in the measured irradiances will cause underestimation of $\tau_{cld}$ by both the RD and RS methods. We correct the $\tau_{cld}$ outputs of both retrievals for errors associated with the FOV of SPN-S. This correction is done by developing an empirical relationship from radiance simulations that associates the magnitude of the irradiance bias induced by the FOV error to the true $\tau_{cld}$ – the details

of the correction are explained in Appendix A.

RD requires measurements of $a_{fl}$, which is derived from flux measurements made by two sensors. The spectral total upwelling irradiance, $F_{up}$, is measured by a nadir mounted Solar Spectral Flux Radiometer (SSFR, Pilewski et al., 2013). SSFR is a moderate resolution total irradiance spectrometer system with a spectral range of 350-2100 nm. Like SPN-S, SSFR is radiometrically calibrated against a NIST traceable lamp standard before and after each field campaign. Throughout the

duration of each campaign, a series of field-calibrations were used to monitor and correct for variations in the primary radiometric calibration. Since the upwelling irradiance is diffuse, the signal is only minorly impacted by the angular response



of the SSFR light collector. A spectrally dependent factor that accounts for the angular response of the SSFR light collector to diffuse irradiance is determined through laboratory investigation. This factor has a value near unity and it is used to correct the upwelling irradiance measurements to account for this angular dependence of the measured signal. $a_{fl}$ is then determined by

ratioing $F_{up}$ to $F_{tot}$, the latter of which is measured by SPN-S.

Altitude, pressure, sun position and navigation data are measured at 1Hz aboard the P-3 as described in Bennett (2020).

### 3.3 Radiative transfer model

The simulated $DR$ values used in RD are made using the Discrete Ordinates Radiative Transfer Program (DISORT) 2.0 (Stamnes et al., 2000). We use the libRadtran library version 2.0.1 to configure and run DISORT 2.0 (Emde et al., 2016), with the base configuration set to:

- o   Molecular absorption is done using LOWTRAN 7, Pierluissi and Pang, (1985).
- o   Pressure, temperature, and gas mixing ratio profiles— including ozone and water vapor —are set using the
Tropical Atmosphere profile from Anderson et al., (1986).
- o   Solar source: 1 nm resolution top of atmosphere flux from Kurucz, (1992).
- o   Slit function with a 6 nm full width at half maximum is used on the output of spectral calculations.
- o   Solar zenith angle and model output elevation were set to values corresponding to P-3 position and time of day.
- o   Flight-level spectral albedo, $a_{fl}$, from the SPN-S/SSFR measurements.


We simulated cirrus clouds by modifying the standard aerosol configuration in libRadtran which generates scattering phase functions from Henyey-Greenstein. A cloud layer was inserted between $10 - 11$km and then we adjusted the layer optical depth to the appropriate $\tau_{cld}$. In this configuration, SSA was set to 1, and $g = 0.85$ was used as a baseline value of the asymmetry parameter. All other simulated irradiances mentioned in the paper, e.g., Figure 1, used a similar model configuration

with deviations noted in the text.

### 3.4 Validation data

The Spectrometer for Sky-Scanning Sun-tracking Atmospheric Research (4STAR, Dunagan et al., 2013) is an airborne sun-photometer that makes direct-beam measurements of $\tau_{aer}$ above the aircraft using spectrometers (similar to

SSFR) spanning 350-1650 nm, with $\tau_{aer}$ reported at 24 wavelengths outside of gas absorption bands. Before and after each field deployment 4STAR is calibrated at Mauna Loa Observatory using Langley extrapolation methods. Additionally, in-flight high-altitude calibration measurements are used as a calibration verification and adjustment throughout deployment (LeBlanc et al., 2020). The resulting uncertainty in 4STAR measured $\tau_{aer}$ is as low as 0.007 at the 501 nm channel.





4STAR operated only during the ORACLES campaign so we also compared the derived $\tau_{aer}$ values from the

retrievals against in situ measurements of optical extinction ($\tau_{insitu}$). For CAMP²Ex, total dry aerosol scattering was measured in situ by a TSI-3563 nephelometer at relative humidity (RH) less than 40%. To account for aerosol humidification, a parallel TSI-3563 nephelometer was operated at an RH of 82 ± 10 % and was used to derive the scattering hygroscopicity factor (i.e., f(RH), Ziemba et al., 2013). Scattering coefficients for each measured wavelength (i.e., 450, 550, and 700 nm) at ambient RH are then calculated using dry scattering coefficients, f(RH), and ambient RH by the diode laser hygrometer (DHL, Diskin et

al., 2002). Optical absorption coefficients were measured by a Radiance Research 3-wavelenght particle soot absorption photometer (PSAP) at 430, 532, and 660 nm, with the sample dried by heating the air to 40°C. To develop altitude profiles of $\tau_{insitu}$, ambient extinction coefficients are computed as the sum of dry absorption coefficient and ambient scattering coefficient, after correcting both to 500 nm wavelength using measured scattering and absorption angstrom exponents and Eq. 6. The estimated uncertainties for the scattering and absorption coefficients are 30% and 15% respectively. For the square

spiral maneuvers performed by the P-3, this total ambient extinction was integrated with respect to altitude to generate a $\tau_{insitu}$ profile. Before integrating, 10 second averaging was applied to the timeseries to reduce the influence of noise artifacts on the profile. At ORACLES TSI-3563 nephelometers and Radiance Research PSAP were also flown, and measurements were processed in the same fashion as those from CAMP²Ex, with a caveat being a pair of Radiance Research M100 nephelometer were humidity controlled and used to determine ambient scattering at 540 nm and f(RH). At ORACLES, the Radiance Research

M100 nephelometer data was used to derive ambient extinction at times when the ambient TSI-3563 nephelometer was not operational. All in situ measurements on the P-3 aircraft for both CAMP²Ex and ORACLES were made behind an isokinetic shrouded solid-diffuser inlet (McNaughton et al., 2007) and are reported at ambient temperature and pressure.

### 3.5 Retrieval implementation

### 3.5.1 Diffuse Method, RD

Implementation of RD is straightforward, and we followed the steps outlined in Section 2.1 of this paper. For the 500 nm SPN-S channel, $DR_{mea}$ is calculated directly from the measured $F_{dif}$ and $F_{tot}$. The estimate of the optical depth, $\tau_{est}$, is made from $DR_{mea}$ using Eq. 5. $\tau_{est}$ is input into the RTM along with $a_{fl}$, to obtain a simulated value of $DR_{sim}$. If $|DR_{mea}/DR_{sim} - 1| \leq 0.01$ the $\tau_{est}$ is the reported $\tau_{cld}$ value of the retrieval. If $DR_{mea}/DR_{sim} > 1.01$, $\tau_{est}$ is increased by

a value of 0.01, and if $DR_{mea}/DR_{sim} < 0.99$, $\tau_{est}$ is decreased by a value of 0.01. The RTM is run again and the new $DR_{sim}$ is compared to $DR_{mea}$. The process iterates until the condition, $|DR_{mea}/DR_{sim} - 1| \leq 0.01$, is met.

We run RD at three wavelengths to obtain $\tau_{cld}$ at 500, 670, and 870 nm. Since cirrus clouds are made of ice crystals, we assume the layer $AE \approx 0$, and therefore the $\tau_{cld}$ at the three wavelengths should be similar if only cloud is present. Significant spectral variation in $\tau_{cld}$ – >5% deviation from the 500 nm retrieval to the 870 nm value – indicates that aerosols

are present in the sampling layer. In such case, careful interpretation of the retrieval output is advised.





### 3.5.2 Spectral Direct Beam Method, RS

The first step in implementing RS is to use Eq. 1 to derive the spectrally dependent total measured optical depth above the aircraft, $\tau_{tot,\lambda}$. The $\mu$ term in Eq. 1 is calculated from the reported solar zenith angles logged in the P-3 flight records. $F_0$
in this calculation is determined using RTM simulations of $F_{dir}$ at 14 km, which is an altitude above the cirrus cloud layer. This is done for all $F_{dir}$ measurements contained within a square spiral where $DR_{mea} < 0.95$, which generally corresponds to $\tau \sim 3$ (see Figure 2). This threshold on the diffuse ratio is used to ensure that there is measurable direct beam signal, and above this level, measurement uncertainty in $F_{dir}$ makes it untenable to use RS. Using the atmospheric pressure measurements made from the P-3, $\tau_{Ray,\lambda}$ is calculated by Eq. 9, and then these values are subtracted from $\tau_{tot,\lambda}$:

$\tau_\lambda = \tau_{tot,\lambda} - \tau_{Ray,\lambda}$                                                      Eq. 10

We do this for a limited set of the SPN-S wavelengths, $\lambda_{retrevial} = \{(460 - 540), (665 - 684), (746 - 755), (772 - 785), (860 - 879)\}$ nm. The wavelengths used in the retrieval were determined by studying simulated spectra of optical depths, such as those in Figure 1, and then selecting spectral regions where there is minimal influence of molecular absorption. The presence of some extinction by trace gases is acceptable at the selected wavelengths because we correct the $\tau_\lambda$ spectra for
these influences using aerosol free samples from the top of the spiral (see the following paragraphs for details of this step). This correction does not account for changes in column trace gas concentrations as the aerosol layer is profiled, which is a limitation of this method. Our analysis was bounded between 460 nm and 880 nm due to the SPN-S calibration issue described in Section 3.2.

The determination of $F_0$, the presence of trace gases, and calibration inconsistencies with the prototype SPN-S system
led to complexities in implementing RS. Flight-to-flight changes in atmospheric conditions and the SPN-S calibration caused the derived $\tau_\lambda$ profiles to have a spurious spectral shape, deviating from flat under clear-sky or cloud only conditions (the faint blue dots in Figure 4 are a good example of errors in $\tau_\lambda$). These errors had a constant magnitude within a flight and were not proportional to signal magnitude. Issues related to the dark current correction may have been responsible for some of the observed behavior, though we are unsure. Additionally, the use of the RTM to derive $F_0$, which is traceable to the solar
spectrum defined by Kurucz (1994), is a source of error in optical depth calculations. Nonetheless, we were able to account for these errors by using $\tau_\lambda$ samples from high-altitude flight to adjust the measured optical depth profiles. To do this, spectral profiles of $\tau_\lambda$ from aerosol free regions of the tops of the P-3 spirals were collected. The spiral tops are assumed to be aerosol free, and this assumption is checked by manually observing if the 4STAR aerosol optical depth ($\tau_{4STAR}$) is < 0.05, or for CAMP²Ex case if the in situ integrated extinction ($\tau_{insitu}$) has a significant gradient in this region of the spiral profile. Spirals
with tops that are within the aerosol layer were excluded from the study. The spectral optical depth correction, $\tau_{crr,\lambda}$, is the mean optical depth that each channel is from a reference channel at 500 nm for the selected spiral top $\tau_\lambda$ spectra:

$\tau_{crr,\lambda} = \overline{\tau_\lambda - \tau_{500nm}}$                                                      Eq.11

The optical depth correction is applied to all $\tau_\lambda$ by subtraction:

$\tau_{adj,\lambda} = \tau_\lambda - \tau_{crr,\lambda}$                                                      Eq.12





Applying this correction is beneficial since Eq. 7 assumes $\tau_\lambda$ has a smooth exponential shape, and correcting $\tau_\lambda$ to a line allows for better representation of the data by the model. Due to this need to correct $\tau_\lambda$, we restrict use of RS to square spiral samples and do not use it for timeseries analysis of full science flights. In a sense, we are deriving layer optical depth with RS, and the spiral tops are used to characterize the flux at the top of the layer and deviations at lower altitude are accounted for by the $\tau_{aer}$ term. RD in comparison, is temporally and spatially independent from itself and can be used at any point along

the flight track when $DR$ is sufficiently small – more details about the utility of both methods are discussed in Section 5. Further, scattering by low- and mid-altitude clouds can interfere with interpretation of $F_{dir}$ and therefore spirals with significant amounts of these clouds were excluded from this study (e.g., if the P-3 entered cloud during the spiral, the spiral was not analyzed).

        Each corrected optical depth profile, $\tau_{adj,\lambda}$, is compared to a series of curves generated using Eq. 7 for defined sets

of $\tau_{cld}$, $\tau_{aer}$ and $AE$. $\tau_{cld}$ is varied from 0 to 5 at 0.01 resolution, $\tau_{aer}$ is varied from 0 to 1.5 with 0.01 resolution, and $AE$ ranged from 1 to 2.0 with 0.1 resolution. The RMSE between each curve and the $\tau_{adj,\lambda}$ sample is computed. The fit parameters- $\tau_{cld}, \tau_{aer}, AE$ -corresponding to the curve with the lowest RMSE value are designated the outputs of the retrieval. Figure 4 shows two examples of RS for a clear-sky case (blue line) and a case within an aerosol layer (red line). The solid filled dots represent $\tau_{adj,\lambda}$ and the impact of $\tau_{crr,\lambda}$ is noticeable when comparing to the faded dots, which are the uncorrected optical

depth profiles, $\tau_\lambda$. The solid lines represent Eq. 7 with the fit parameters set to the retrieval output in these two cases.

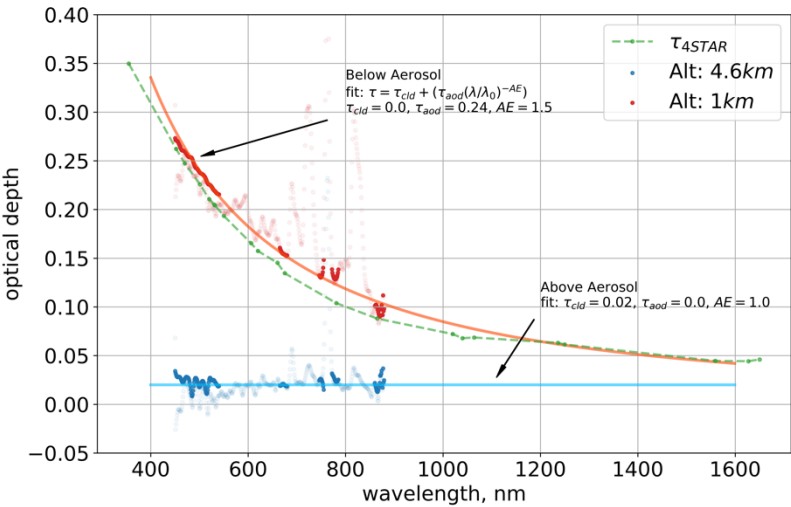

**Figure 4:** Two examples of the spectral optical depth used in RS to determine $\tau_{cld}$ and $\tau_{aer}$. Both examples are from the spiral on 20181005 of the ORACLES mission. (1) A clear-sky case, colored in blue, from a sample taken at an altitude 4.6 km. (2) In red, a sample from 1 km which is within the aerosol layer. The dark red/blue dots are





$\tau_{adj,\lambda}$, and they correspond to the red/blue dashed lines in Figure 1. The faded red/blue dots are $\tau_\lambda = \tau_{tot,\lambda} - \tau_{Ray,\lambda}$ values from all SPN-S channels from 460-880 nm, including those not used in the retrieval. The blue and red lines represent the best-fit lines, for the RS outputs. The green line is $\tau_{4STAR}$ as measured at 1 km altitude.

## 4 Results

We first present the results of the ORACLES campaign followed by the CAMP2Ex results. For each campaign we first show the retrieval performance for a one square-spiral case study, and then we give the aggregate statistics for the campaign.

### 4.1 ORACLES OCT 3, 2018 profile

Between 09:52:00 and 10:15:00 UTC, the P-3 flew a square spiral centered around a latitude and longitude of -6°35'36.60", 6°58'12.36", with a spiral midpoint SZA of 22.8°. The profile started at an altitude of 5.74 km and ended near the surface at 0.33 km. No clouds were present above the spiral start height, and this absence of cirrus was typical of the spirals conducted during the 2018 deployment of the ORACLES campaign. Despite this lack of cloud, we applied RD in addition to RS to compare the outputs of the two methods. The left panel of Figure 5 shows both the measured 500 nm diffuse ratio, $DR_{mea}^{500nm}$, and 500 nm direct transmittance, $T^{500nm}$ with their associated uncertainty. Starting just below the 4 km level, the magnitude of the slopes of both $DR_{mea}^{500nm}$ and $T^{500nm}$ steepen signaling the presence of an aerosol layer. The profiles remain predictable in shape all the way to the surface, with monotonic change, and no erratic points to indicate clouds along the profile path.

The right panel of Figure 5 shows the altitude profiles of 500 nm $\tau_{aer}$ from RS ($\tau_{RS,aer}$, green), 4STAR ($\tau_{4STAR}$, red), in situ ($\tau_{insitu}$, cyan), and RD ($\tau_{RD}$, blue). Expectedly, the inability of RD to account for absorption within the aerosol layer leads to lower values of $\tau_{aer}$ than the other methods below 3 km. Above the aerosol layer, RD detects an optical depth of ~0.05, and this is similar to 4STAR, which commonly measures non-zero optical depths at the top of the spiral. These optical depths are mostly due to the presence of stratospheric aerosols (Kremser et al., 2016), though there may also be contribution from small amounts tropospheric aerosol as well. In the case of 4STAR, the $\tau_{aer}$ measured at the top of the spiral is accounted for by subtracting off a constant value that is equal to the mean $\tau_{4STAR}$ value above 5.5 km. Visual comparison between $\tau_{RS,aer}$ and $\tau_{4STAR}$ profiles show agreement with $\tau_{4STAR}$ falling within the reported uncertainty of $\tau_{RS,aer}$ for all samples outside of a few $\tau_{4STAR}$ outlier cases. With respect to $\tau_{insitu}$ profile, both the $\tau_{RS,aer}$ and $\tau_{4STAR}$ agree well with the in situ measured extinction.





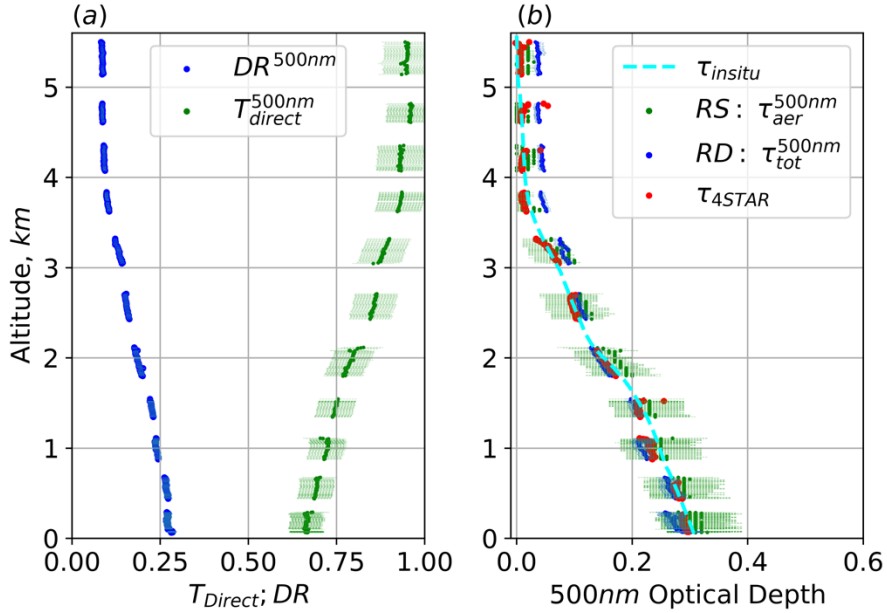

**Figure 5:** ORACLES spiral on 20181003. (a) $DR_{mea}^{500nm}$ and $T^{500nm}$ with the associated measurement uncertainty shown by the shading. (b) $\tau_{aer}$ profiles derived from RS, 4STAR, in situ, and RD.

## 4.2 2018 ORACLES mission statistics

There were 14 square spirals flown by the P-3 during the 2018 deployment of the ORACLES mission. The cumulated $\tau_{aer}$ data from this set of spirals is used in regression analysis with 4STAR serving as the reference measurement. The left panel of Figure 6 shows scatter plots of $\tau_{4STAR}$ vs $\tau_{RS,aer}$ and $\tau_{insitu}$. Regression for $\tau_{RS,aer}$ gives $R^2 = 0.96$, slope = 0.96, intercept = $1.4 \times 10^{-3}$, and RMSE is $3.0 \times 10^{-2}$. The high $R^2$ value and low RMSE indicate that the SPN-S based aerosol optical depth retrieval preforms well in comparison to 4STAR. Regression for $\tau_{insitu}$ vs $\tau_{4STAR}$ yields $R^2 = 0.96$, slope = 0.90, intercept = $-2.1 \times 10^{-4}$ and RMSE is $3.3 \times 10^{-2}$. The good agreement between $\tau_{4STAR}$ and $\tau_{insitu}$ gives us confidence that integrating the in situ measurements of extinction can serve as a basis for comparison for the CAMP$^2$Ex cases where 4STAR (or an equivalent system) was not operated. The right panel shows percent errors for $\tau_{RS,aer}$ and $\tau_{insitu}$ for binned $\tau_{4STAR}$ values. At small optical depths, uncertainty is a significant fraction of the total optical depth signal, and this leads to large relative errors. As $\tau_{4STAR}$ increases, the fractional error between $\tau_{4STAR}$ and $\tau_{RS,aer}$ decreases.





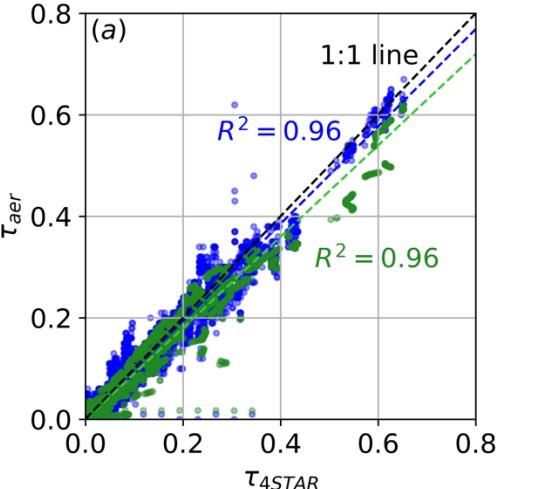
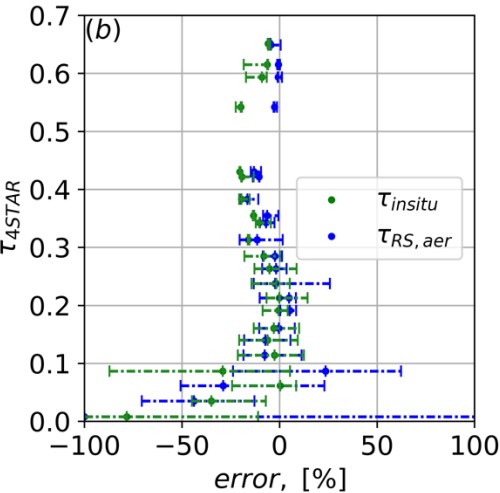

**Figure 6:** Aggregated $\tau_{aer}$ statistics using $\tau_{4STAR}$ as a baseline for the ORACLES campaign. (a) scatter plots of $\tau_{4STAR}$ vs $\tau_{RS,aer}$ and $\tau_{insitu}$. Regression or $\tau_{RS,aer}$ gives $R^2 = 0.96$, slope = 0.96, intercept = $1.4 \times 10^{-3}$, and RMSE is $3.0 \times 10^{-2}$. Regression for $\tau_{insitu}$ yields $R^2 = 0.96$, slope = 0.90, intercept = $-2.1 \times 10^{-4}$ and RMSE is $3.3 \times 10^{-2}$. (b) Percent error in $\tau_{RS,aer}$ and $\tau_{insitu}$ relative to $\tau_{4STAR}$. $\tau_{4STAR}$ values are sorted into bins with 0.25 optical depth width, and the corresponding statistics for $\tau_{RS,aer}$ and $\tau_{insitu}$ are reported. Dots represent the median value; tips of line are the 25 and 75 quantiles of the $\tau_{RS,aer}$ and $\tau_{insitu}$ distributions within each bin.

Figure 7 compares the total optical depths derived using RS and RD, where $\tau_{RS,tot} = \tau_{RS,aer} + \tau_{RS,cld}$. For these ORACLES cases conducted under cloud-free skies, the retrieval of $\tau_{RD}$ is caused by aerosols. The regression slope (slope = 0.84) deviates significantly from unity, which is expected because RD accounts for extinction only due to scattering. From our analysis in Section 2.3 we anticipate the regression to give a slope of less than one, with the value being linked to $g$ and SSA of the aerosol layer (the green line in Figure 7 is the modeled relationship for $g = 0.85$ and SSA = 1). The aerosols sampled at ORACLES were absorbing, with mid-visible SSA values near 0.85 (Cochrane et al., 2020), and this sink of radiation by the aerosol layer, along with errors in the scattering phase function used in RD, are the main causes for $\tau$ being underestimated by the RD method. A final point worth noting is that there are numerous instances when $\tau_{RD}$ has a non-zero value yet $\tau_{RS,tot}$ does not register an optical depth. This is not surprising given the higher sensitivity to small changes in $\tau$ of the RD output.





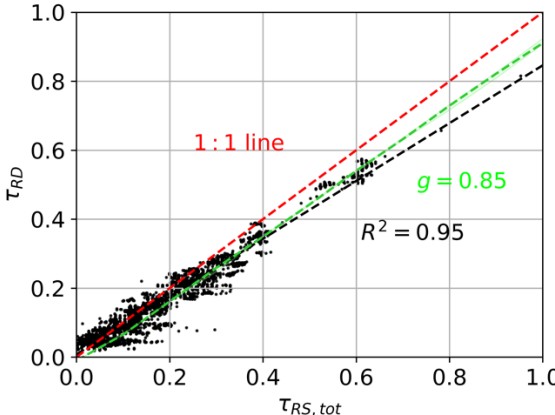

**Figure 7:** Aggregated optical depth data from ORACLES campaign $\tau_{RS,tot}$ vs $\tau_{RD}$. Regression gives $R^2 = 0.95$, slope $= 0.84$, intercept $= 7.9 \times 10^{-3}$. The green line is the predicted $\tau_{RD}$ for a given $\tau_{RS,tot}$ value, which is determined by RTM simulations (i.e., the red dashed line in the center panel of Figure 3), for $g = 0.85$.

### 4.3 CAMP²Ex September 16, 2019 profile

From 00:57:00 to 01:27:00 UTC on 20190917 the P-3 flew a square spiral centered at latitude and longitude of
13°55'52.32", 125°27'41.76", with a spiral midpoint SZA of 36.1°. The spiral profiled from an altitude of 5.59 km to the near the surface at 0.40 km. In contrast to the ORACLES case study, there were cirrus clouds present above the spiral location and their radiative signature can be seen in the deviation of both the $DR_{mea}^{500nm}$ and $T^{500nm}$ profiles from a monotonic curve above 3 km (left panel of Figure 8). Additionally, the dip in $T^{500nm}$ value (seen as a spike in $DR_{mea}^{500nm}$) around 2 km indicates the influence of a cloud in the vicinity of the P-3. The variation in $DR_{mea}^{500nm}$ and $T^{500nm}$ due to clouds causes difficulty when
visually attempting to discern the start of aerosol layer on the graph, but near 2.5 km the magnitudes of the $T^{500nm}$ and $DR_{mea}^{500nm}$ slopes increase indicating the presence of aerosols.

The middle panel of Figure 8 shows the altitude profiles of $\tau_{RS,aer}$ and $\tau_{insitu}$. There is good agreement between the shape of the two curves, however $\tau_{RS,aer}$ is consistently higher than $\tau_{insitu}$ for altitudes below 2.5 km. We are not sure of the specific reasons why $\tau_{RS,aer}$ has higher values than $\tau_{insitu}$ in this case. This bias was not found regularly in the other CAMP²Ex
cases which indicates that instrument error, associated with either SPN-S or the in situ measurements, may be responsible for the observed differences in $\tau_{aer}$.

The right panel of Figure 8 shows cloud optical depth derived from the RS and RD methods, $\tau_{RS,cld}$ and $\tau_{RD}$ respectively, as the P-3 profiled the atmosphere. $\tau_{RD}$ is shown at 500, 670 and 870 nm. Above the aerosol layer (>2.5 km) there is good consistency between the three wavelengths of $\tau_{RD}$. The 870 nm channel has slightly greater variance than the
other two and this relates to increased noise levels and stray-light issues within the spectrometer that we observed for SPN-S channels past 850 nm. Above the aerosol layer $\tau_{RD}$ has less variance in optical depth values than $\tau_{RS,cld}$, though the majority





of samples of $\tau_{RS,cld}$ and $\tau_{RD}$ fall within their uncertainties. An imperfect attitude correction of the SPN-S appears to be the main driver of this discrepancy between the retrieved $\tau_{RS,cld}$ and $\tau_{RD}$ values. Since fluctuations in sensor attitude more severely impact the direct irradiance than the diffuse ratio, changes in the P-3 heading during the square spiral will influence $\tau_{RS,cld}$

more than $\tau_{RD}$. One benefit of having two methods for retrieving $\tau_{cld}$ is that sources of error that are a result of the experimental setup can be identified. We do not account for the uncertainty attributed to changes in sensor attitude on RS because there are engineering approaches, such as stabilizing platforms, and study methods that can be implemented which can reduce the impact aircraft attitude has on the measurement in future deployments of SPN-S.

Within the aerosol layer (<2.5 km), spectrally dependent absorption of the aerosol causes the three wavelengths of

$\tau_{RD}$ to diverge, with the longer wavelengths less influenced by the aerosols. While RD is limited when sampling in the aerosol layer because SSA is not known, the wavelength dependence of $\tau_{RD}$ can be used to determine if aerosols are present. For $\tau_{RS,cld}$, the retrieved values from within the aerosol layer fall within the range of cloud optical depth observed above the aerosol layer.

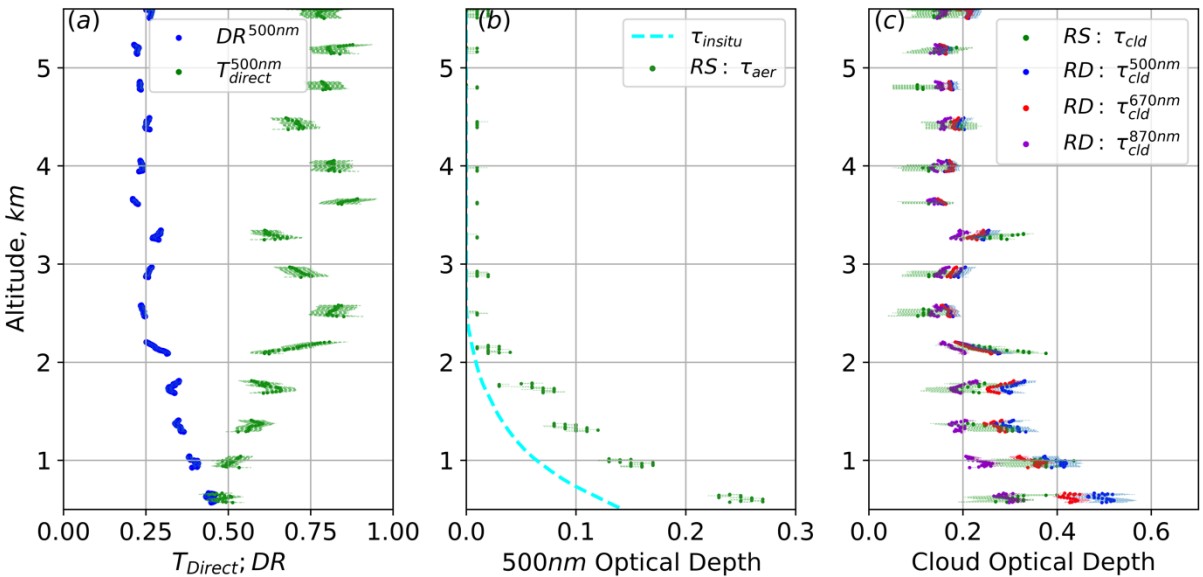

**Figure 8:** CAMP²Ex square spiral on 20190916. (a) $DR_{mea}^{500nm}$ and $T^{500nm}$ as a function of P-3 altitude with the associated measurement uncertainty shown by the shading. (b) $\tau_{aer}$ profiles derived from RS and in situ. (c) $\tau_{cld}$ values for RD at 500, 670, 870 nm and RS as a function of altitude.






**4.4 2019 CAMP2Ex mission statistics**

There were 18 spirals flown by the P-3 during the CAMP$^2$Ex campaign. Figure 9 shows the relationship of $\tau_{RS,aer}$ vs
$\tau_{insitu}$ and the corresponding regression gives $R^2 = 0.97$, slope = 0.94, intercept = 2.4×10$^{-4}$, while RMSE is 3.4×10$^{-2}$. These
aggregated results are consistent with the comparison between $\tau_{4STAR}$ and $\tau_{insitu}$ done for the ORACLES campaign, where
$\tau_{insitu}$ also had a slight low bias in relation to the 4STAR derived optical depths (slope = 0.90). The relatively clean air with
lower aerosol extinction and more varied source regions sampled during the CAMP$^2$Ex campaign (Hilario et al., 2021) restrict
our ability to fully validate $\tau_{RS,aer}$ under the cirrus conditions because the amount to sampling done at high $\tau_{aer}$ was limited.
Additional future work to further examine the retrieval of $\tau_{aer}$ under a broader scatter range of optical depths when cirrus clouds are
present is needed. This is critical in light of the found limitations of the SDA method under cirrus conditions (Smirnov et al.,
2018). However, we expect that since $\tau_{RS,aer}$ is a layer, and not a column optical depth, the forward scattering of light by cirrus
which inhibits the SDA method to be less troublesome for our RS method because the irradiance at the top of the layer is
directly characterized. Regardless, the limited retrieved $\tau_{RS,aer}$ values at CAMP$^2$Ex are consistent with relationship between
of $\tau_{insitu}$ and $\tau_{4STAR}$ observed at ORACLES.

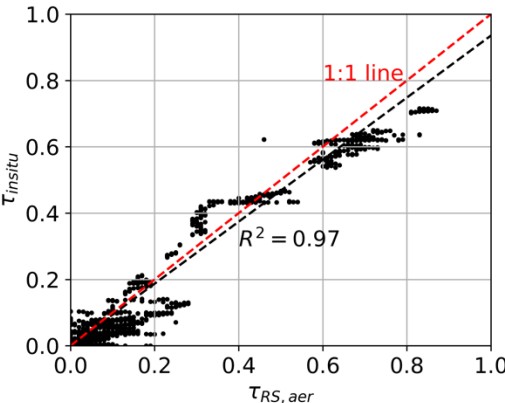

**Figure 9:** $\tau_{RS,aer}$ vs $\tau_{insitu}$ comparison from the CAMP$^2$Ex campaign.

Like with the ORACLES results, we compare $\tau_{RS,tot}$ to $\tau_{RD}$ for two cases: (1) above the aerosol layer, which is shown
in the left panel of Figure 10, and (2) the optical depths for complete spiral profiles (i.e., data from above and within the aerosol
layer), is shown in the right panel of Figure 10. At higher optical depths, the relationship between $\tau_{RS,cld}$ and $\tau_{RD}$ (left panel)
resembles the predicted one (see Section 2.3), with the sampled points clustering mostly along the 1:1 line. There is a grouping
of points near $\tau_{RS,cld} = 1$, that have lower values of $\tau_{RD}$, which are depressing the regression slope from unity (slope = 0.94).
A possible explanation for these points is clouds in the vicinity of the P-3 scattering light into the diffuse sensor of the SPN-
S, biasing $F_{dif}$ high, the result of which is low $\tau_{RD,cld}$. Comparing the relationship between the total set of samples ($\tau_{RS,tot}$ vs


$\tau_{RS}$, right panel), the points generally falling below the 1:1 line (slope = 0.89). Here, the addition of absorbing aerosols is the likely cause of the low $\tau_{RD}$ values in relation to $\tau_{RS,tot}$. However, errors in the scattering phase function used in the RTM and the influence of mid-spiral clouds may also be partially responsible for the lower values of $\tau_{RD}$ and the resulting low regression slope.

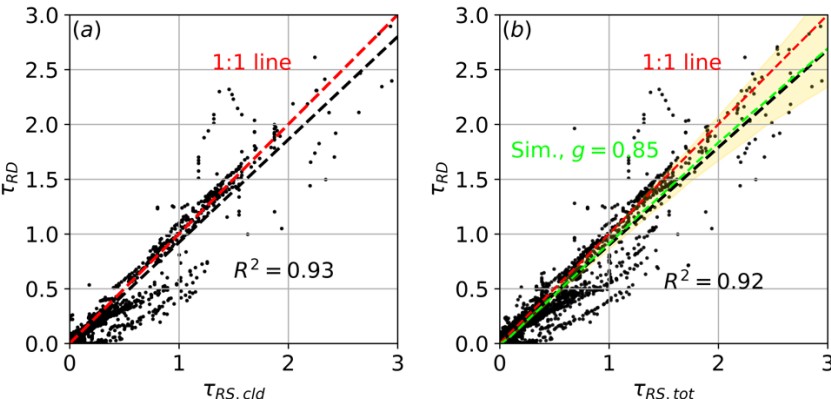

**Figure 10:** CAMP²Ex: (a) Above aerosol layer comparison: $\tau_{RS,cld}$ vs $\tau_{RD}$. (b) All data total optical depth comparison: $\tau_{RS,tot}$ vs $\tau_{RD}$. The dashed green line represents the expected $\tau_{RD}$ given a value of $\tau_{RS,tot}$ assuming $g = 0.85$.


### 4.5 RD cirrus characterization along flight track

The inputs of RD are absolute, and therefore the method can be deployed to derive $\tau_{cld}$ along P-3 flight tracks. We demonstrate this with an example from the science flight on 20190929 from UTC 03:00-05:00, with the 1 Hz SPN-S and SSFR irradiance data is subsampled at 0.1 Hz before being processed. For this sampling period SZA ranged from 10.0° to 17.6°.

The top panel of Figure 11 shows the P-3 altitude and $DR_{mea}^{500nm}$, while $\tau_{cld}$ at 500 and 670 nm are shown in the bottom panel, along with the ratio of the two optical depths. The presented data are not filtered for any criteria on $DR$. When $DR$ is near unity the retrieval preforms poorly, resulting in spikes in the retrieved value of $\tau_{cld}$ – two examples of these events are seen at 03:30, and 04:30. When $DR < 0.9$, such as the start of the flight track from 03:00-03:28 or the section near the end, 04:45-04:56, there is consistency between $\tau_{cld}^{500nm}$ and $\tau_{cld}^{670nm}$, indicating successful retrieval of $\tau_{cld}$. At lower altitudes, from 03:38-04:15,

the cloudier environment frequently causes the irradiance to become completely diffuse and the retrieval fails. During the low flight leg, 04:00-04:10, the ratio of $\tau_{cld}^{500nm}/\tau_{cld}^{670nm}$ is high, indicating the possible presence of aerosols. A complete analysis of CAMP²Ex cirrus cloud optical and radiative properties is provided in Hong et al., (2021).



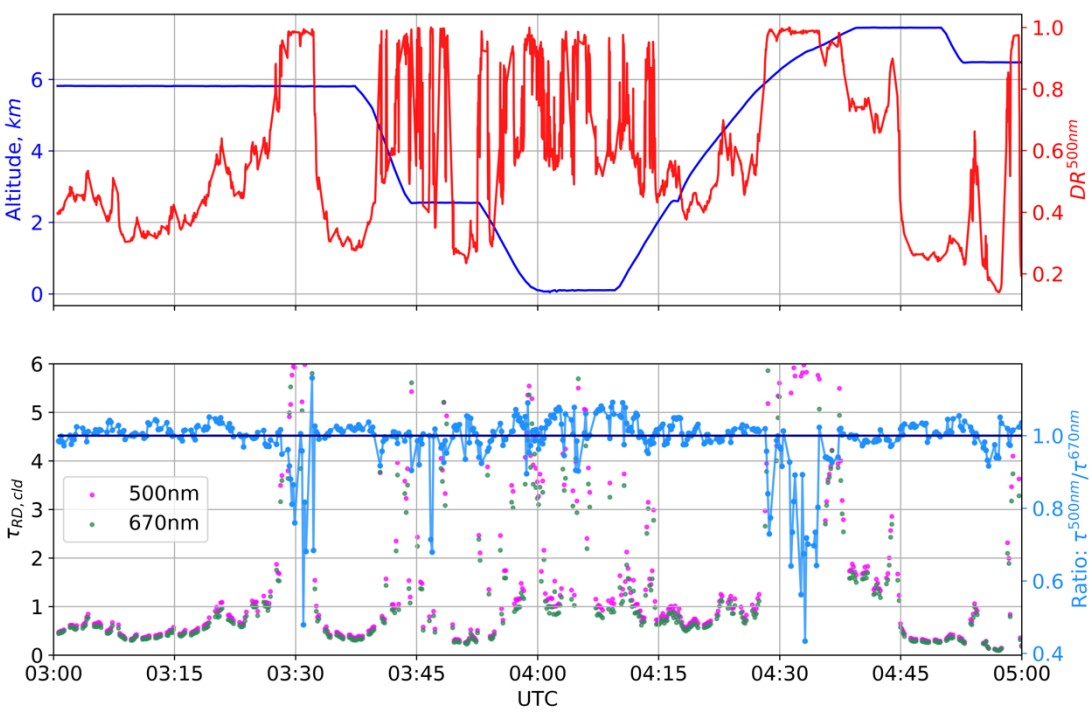

**Figure 11:** Time series of 20190929 partial flight-track of (a) P-3 Altitude and $DR_{mea}^{500nm}$, (b) $\tau_{RD,cld}^{500nm}$ and $\tau_{RD,cld}^{500nm}$. The ratio, $\tau_{RD,cld}^{500nm}/\tau_{RD,cld}^{670nm}$ is also shown.

## 5 Discussion

The application of the SPN-S radiometer to deriving overlying cloud and aerosol optical depth is promising, but there are tradeoffs that must be considered when comparing these methods to existing standards. If the objective is to identify and classify the optical properties of thin cloud (e.g., $\tau_{cld} < 1$), RD is a robust choice because the small uncertainty in $DR$ allows for a highly sensitive retrieval. At optical depths greater than 1, the assumptions underpinning RD, especially knowledge of

the scattering phase function, lose validity, causing significant errors in the retrieval output (see Figure 3). Since the relationship between the retrieved $\tau_{cld}$ and its error can be largely explained for a given value of $g$, it may be possible to correct for the discrepancy between predicted and true cloud optical depth under conditions where $g$ is constrained or there is accurate knowledge of the phase function. However, the capabilities of RD will always be limited to thin clouds because beyond optical depth of about 5 the $DR$ signal loses much of its sensitivity to changes in $\tau_{cld}$. Of course, this limit at which RD can accurately

derive $\tau_{cld}$ will be dictated by the performance of the sensor measuring $DR$. Another limitation of RD is that, without knowledge of the layer SSA, the effects of absorption on $DR$ are not quantifiable, thus making the retrieval of limited use when





aerosols are present. The caveat to this is the development of a simple aerosol flag, where spectral dependencies in RD output can be attributed to an absorber, such as aerosols, being present in the sampled layer.

The benefit of RS is that the retrieval has the potential to separate the cloud and aerosol radiative signals from each other. The tradeoff with the existing sun-photometry standards of measuring $\tau_{aer}$ is greater uncertainty that stems from the difficulty in accurately measuring $F_{dir}$ with a radiometer. Errors are induced into $F_{dir}$ measurements by several mechanisms: changes in sensor attitude, calibration shifts over time, cosine response errors, temperature effects, etc. We attempt to reduce the uncertainty associated with these errors by sampling profiles of the atmosphere and using the high-altitude aerosol-free samples to correct for calibration errors and variations in atmospheric composition, but the need for this correction currently limits the application of the method to profiles of atmospheric layers. While it is unlikely that the SPN-S system will ever be able to obtain the precision and accuracy of a system like 4STAR, there remains obvious room for improvement. Better characterization of the offset angles of the mounted SPN-S will reduce the error related to changing aircraft attitude. (During ORACLES and CAMP²Ex, SPN-S was not a priority instrument and flying the necessary flight patterns to determine the offset angles via the Long et al, (2010) method was not possible.) More sophisticated radiometric calibrations will also improve retrieval performance, be it through laboratory comparisons with traceable lamp standards or using Langley techniques and intercomparison with known benchmarks. The 7-detector head design of SPN-S introduces technical challenges when calibrating using a lamp, and this being a prototype instrument, some of the challenges are still being worked through.

Another source of uncertainty related the SPN-S is the wider FOV shadow band radiometer systems have compared to sun-photometers ($\sim 5 - 10°$ vs $\sim 1°$ ) which results in an overestimation of the direct transmittance (di Sarra et al, 2015; Wood et al, 2017). Empirical methods can be used to account for the discrepancy in sensor FOV, but these methods are reliant on large amounts of co-located measurements of $\tau_{aer}$ with a sun-photometer under varying aerosol or cloud conditions. These empirical relationships are specific to the individual sensors themselves, meaning that we cannot apply previously generated FOV corrections here. In the specific cases used in this study, comparison with SPN-S and 4STAR did not indicate substantial FOV bias that warranted correcting $\tau_{aer}$ so we did not apply those techniques. However, forward scattering in the direction of the direct beam is most severe under thin ice clouds and so we used RTM simulations of the diffuse radiance to account for these effects when retrieving $\tau_{cld}$ using both RS and RD (details contained in Appendix A).

RS is prone to errors when differentiating a signal between $\tau_{cld}$ and $\tau_{aer}$ if certain aerosol types are present. Coarse mode aerosols, such as dust particles, can have $AE$ values near zero (Eck, et al., 1999), and therefore have minimal spectral dependence to their optical depth. That is, larger aerosol particles have extinction wavelength profiles similar to cirrus clouds. In cases where large aerosol particles are present, the $\tau_{cld}$ term in Eq. 7 is also dependent on the coarse-mode aerosol optical depth. The aerosol type and clouds present when sampling will determine the extent to which large aerosols impact the retrieval output. For example, if the study region is cloud free, optical depths due to coarse-mode aerosols will still be able to be measured using RS by evaluating the $\tau_{cld}$ term of the retrieval output. In fact, RS could be extended to do aerosol mode analysis under cloud-free skies in a manner similar to the SDA method used in sun-photometry (O'Neill et al., 2003).



Advances in the capability of SPN-S are ongoing and improvements to the system have already been made since the prototype version was deployed to ORACLES and CAMP²Ex. Most notably, software improvements now allow for sequential sampling to occur at multiple spectrometer integration times. The advantage of this technique is that both bright and dark parts of the irradiance spectrum can be resolved nearly simultaneously, giving much better instrument performance at the tails of its spectral range (<450 nm and >900 nm). Having greater spectral range in which to evaluate Eq. 7 will allow differentiating

between cloud and aerosol optical depths to be done with greater confidence. Likewise, with the improved calibration mentioned earlier in this section it may be possible to eliminate the need for spiral-patterns and the associated optical depth correction. Instrument upgrades, along with measuring and accounting for extinction from trace gases (i.e., better representation of $\tau_{mol,\lambda}$), opens the door for applying RS to a broader set of sampling types, such as along full flight tracks or to ground-based deployments. Making the mentioned improvements to SPN-S and using it in ground-based settings would

produce useful data to use in the characterization of the retrieval uncertainty. All of these retrieval methods need more future work to better validate and understand their outputs, some of which is currently ongoing. Hong et al., (2021) will compare retrieved $\tau_{cld}$ to similar results from space-borne remote sensors.

        There are several more advantages of SPN-S system-based methods for cirrus cloud and aerosol studies to note. The SPN-S is a total-diffuse radiometer that has no moving parts which makes it an inexpensive and user-friendly instrument to

operate in a wide variety of settings. Airborne sun-photometer or lidar systems tend to be mechanically and technologically complex resulting in significant overhead when operating them in field settings. Moreover, while the spectral analysis techniques of RS can be applied to any set of spectral direct irradiance measurements (for example, RS can be applied to spectral sun-photometer irradiances), a well characterized SPN-S system has inherent advantages. The SPN-S measurement of hyperspectral total irradiance provides a more complete view of the radiative environment rather than informing about only

the optical depth and direct irradiance that sun-staring sun-photometry measurements provide. An example of how this might be beneficial is for the determination of heating rates in the atmosphere. With a measured spectral $\tau_{aer}$ profile, radiometric based approaches have been advanced that allow aerosol intensive properties to be derived (Cochrane, 2020), from which it is possible to determine aerosol heating rates and radiative effects (Cochrane, 2021). SPN-S measurements may allow for similar studies to be completed with a consolidated set of instruments at lower cost. Additionally, aerosol intensive properties can be

studied using the DR measurement in manners similar to Kassianov et al. (2007).

**6 Conclusion**

        In this paper we used the capabilities of the newly developed SPN-S radiometer to implement two retrieval methods: RD is a scheme that utilizes single channel measurements of $DR$ and $a_{fl}$ to derive cirrus cloud optical depth, while RS is a

technique that exploits structure in optical depth spectrum to partition it into $\tau_{cld}$ and $\tau_{aer}$ components. Since the primary radiometric input of RD is the ratio of two measured irradiances, calibration-induced uncertainties in the system are minimized, resulting in a retrieval that is highly sensitive to small optical depths. Unquantified absorption in the atmosphere limits the utility of RD to derive $\tau_{cld}$, with the caveat that the method can be used to identify the presence of aerosols by comparison of


the retrieved optical depths at multiple wavelengths. On the other hand, RS is based on measurements of $F_{dir}$ which have
larger associated uncertainties stemming from calibration errors and the influence of changing sensor attitude. This makes RS
best suited for deriving $\tau_{cld}$ at values greater than unity where the inherent retrieval errors are a lower fraction of the output,
or when aerosols are present. Since $\tau_{aer}$ is derived from the spectral shape of the optical depth, and not the absolute value at
any one wavelength, the $\tau_{aer}$ output of the retrieval is less influenced by the measurement uncertainty than the spectrally-
independent $\tau_{cld}$ output.

We apply both methods to data from two field campaigns, ORACLES in 2018 and CAMP²Ex in 2019, to evaluate
their performance. RS performed well at retrieving $\tau_{aer}$ by comparison to measurements made by the 4STAR sun-photometer
system (RMSE = $3.0 \times 10^{-2}$) and optical depth as retrieved by in situ measurements both under clear-sky and cirrus conditions
(RMSE = $3.4 \times 10^{-2}$). There were limited cases of high aerosol loading under cirrus conditions so the retrieval performance
under these circumstances warrant further investigation. The $\tau_{cld}$ retrievals of both methods were evaluated against each other
and behaved as our theoretical analysis predicted.

The optical depth retrieval uncertainties of these two new methods suggest that SPN-S is not a replacement for
traditional sun-photometer instruments such as 4STAR. However, it is a low-cost alternative that is mechanically simple,
making it logistically easier to deploy in many circumstances, such as on aircraft. Depending on the experiment, the tradeoffs
in optical depth uncertainty of the SPN-S can be afforded by the accessibility it provides to reliably identifying the presence
of above-aircraft clouds and aerosols. Cirrus identification is of value to other passive nadir-viewing sensors such as imaging
and scanning radiometers and polarimeters. SPN-S is also advantageous in that it measures spectrally-resolved total, direct and
diffuse irradiance which are useful quantities in the context of radiation science.

The SPN-S used in this study is a prototype, and work remains to better characterize the performance and calibration
characteristics of the instrument. Much of this work is ongoing already, with software advancements having expanded the
spectral range of the system, while improvements to the calibration procedures are a focus of current work. Increases in system
predictability and reduced measurement uncertainty will allow for more versatility in the deployment of the methods presented
in this study. For example, it may be possible to use RS for timeseries analysis if the calibration stability gets to the point
where a profiling approach is no longer needed to correct for errors in the optical depth spectra. The combination of utility,
robustness and ease of implementation offered by the SPN-S make it feasible to implement in a wide variety of settings: from
future airborne campaigns to long-term monitoring applications at ground-based field sites.

## Appendix A: FOV correction development

Forward scattering of radiation in the direction of the direct beam is a common phenomenon sun-photometry
techniques must account for when deriving optical depth of aerosols and thin clouds (Segal-Rosenheimer et al., 2013).
Frequently the full-angle FOV (here FOV refers to the full-, not half-angle, FOV of the sensor) of a sensor used to measure
$F_{dir}$ is greater than the angular width of the Sun's disk, causing diffuse light surrounding the direct beam to influence
measurements. This overestimation of the direct transmittance leads to $\tau$ being underestimated when implementing standard



radiative transfer techniques. While the SPN-S does not function in the same manner as a sun-photometer to measure $F_{dir}$, the shadow mask used to block the direct beam when sampling $F_{dif}$ has a shading area wider than the beam. That is, the shading

area of the shadow mask is too large, and this leads to a low bias in measured $F_{dif}$ and a high bias in derived $F_{dir}$. Under most atmospheric conditions this bias is minimal and can be ignored. However, it has been shown that when thin clouds are present, especially ice phase clouds which scatter strongly in the direction of the direct beam, the biases can lead to errors in $\tau$ retrievals up to 100 percent (Segal-Rosenheimer et al., 2013).

To account for the contamination in measured $F_{dir}$ by diffuse radiation, we developed corrections for the $\tau_{cld}$ outputs

of both methods presented in this paper, RS and RD, based on simulated radiance fields in the FOV of the SPN-S. These corrections are in the form of a relationship between the optical depth inferred by the sensor to the true optical depth, and they are dependent on solar zenith angle and wavelength.

To do this, we used the Mystic Monte Carlo Model that is a part of the libRadtran package (Emde et al., 2016) to simulate the diffuse radiance, $L_{dif}$, in the FOV of SPN-S. SPN-S is a prototype and its exact FOV is unknown, but it is

estimated to be around $8^o$ (Wood et al., 2017). Further, errors in FOV at angles past $5^o$ will have minimal impact the optical depth correction because the majority of $L_{dif}$ is found within the first angular degree or two from the center of the solar disk. We run a set of radiance simulations, with the sensor pointed towards the Sun's position, and calculate $L_{dif}$ across the arc length of the FOV – scanning across the sensor FOV area in the azimuth and zenith directions. These simulations are done for a set of ice clouds with $\tau_{cld}$ ranging from 0-6 at 0.1 resolution, over a black surface. We limited the Monte Carlo model runs

to 1000 photons each, which is a low amount, but the FOV corrections are based on fits to sets of model runs and so the error of any one simulation due to the low photon count is minimal. Figure A1 shows an example of simulated diffuse transmitted radiance, defined as $Tl_{dif} = L_{dif}/F_0$, as a function of viewing angle in the azimuth. Forward scattering from thin cirrus peaks around $\tau_{cld} = 1$, leading to the observable spike in $Tl_{dif}$.



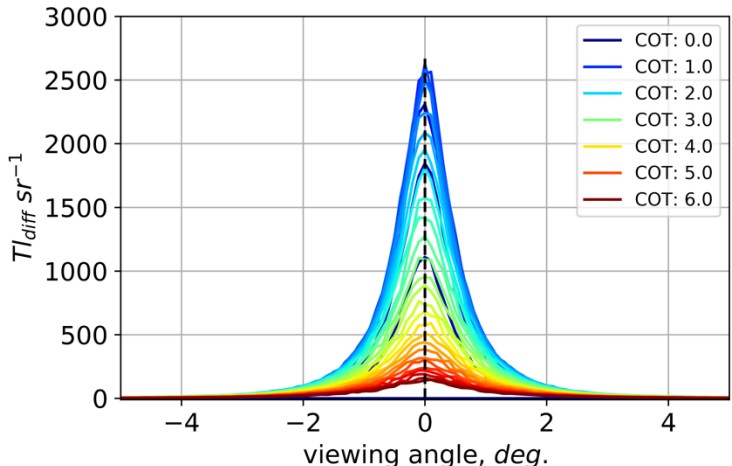

**Figure A1:** Simulations of diffuse radiance transmittance at 500 nm, $Tl_{dif} = L_{dif}/F_0$ when scanning across the sensor FOV in the azimuthal direction. Line color corresponds to $\tau_{cld}$ value. The Sun is positioned at a viewing angle of $0^o$, and the SZA is $20^o$.


We assume that the radiance field is symmetrical about both the zenith and azimuth axis and integrate the radiance fields to derive the total diffuse transmittance in the FOV of the sensor, $T_{dif,FOV}$:

$$T_{dif,FOV} = \iint_0^{fov} Tr_{dif} \, d\theta d\phi \qquad\qquad \text{Eq. A1}$$

In practice, we split the integral along the four FOV paths we simulated using Mystic:

$T_{dif,FOV} = \frac{\pi}{2}\left[\int_{0^o}^{4^o} Tl_{dif} \, d\phi + \int_{-4^o}^{0^o} Tl_{dif} \, d\phi + \int_{0^o}^{4^o} Tl_{dif} \, d\theta + \int_{-4^o}^{0^o} Tl_{dif} \, d\theta\right]$    Eq. A2

$T_{dif,FOV}$ is the extra transmittance in the measurement of direct beam transmittance made by the sensor. Figure A2 shows the dependence $T_{dif,FOV}$ has on $\tau_{cld}$.



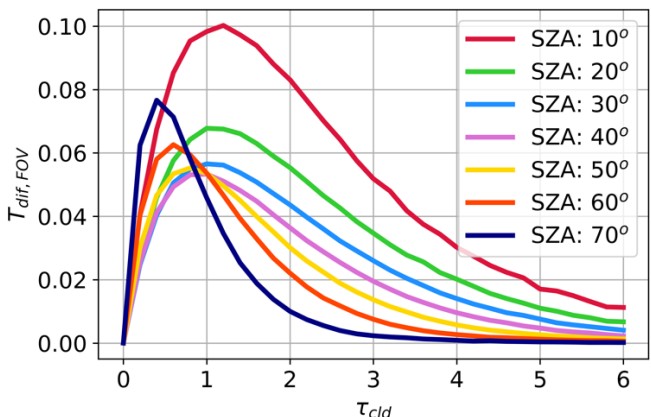

**Figure A2:** $T_{dif,FOV}$ as a function of $\tau_{cld}$ for the 500 nm wavelength. The relationship is shown for multiple SZA ranging from $10 - 70^o$.

For RS, the relationship between $\tau_{cld}$ and the optical depth inferred from the method, $\tau_i$, is determined by inverting Beer's
Law (Eq.1) and inserting the biased transmittance:

$$\tau_i = -\mu ln(T_{dir,true} + T_{dif,FOV}) \qquad \text{Eq. A3}$$

where $T_{dir,true} = T_{dir,true}(\tau_{cld})$, is the direct beam transmittance given a true cloud optical depth. Here again we use libRadtran to simulate $T_{dir,true}$ for each $\tau_{cld}$ value used in the simulations of $T_{dif,FOV}$. Likewise, for RD, $T_{dif,FOV}$ is used to calculate the bias in $DR$ and then Eq. 5 is used to find $\tau_i$:

$\tau_i = \mu ln\left(1 - \dfrac{F_{dif,true} + F_{dif,FOV}}{F_{tot,true}}\right) \qquad \text{Eq. A4}$

where $F_{dif,true}$ and $F_{tot,true}$ are the true irradiances given a true cloud optical depth, and $F_{dif,FOV}$ is the extra diffuse irradiance the sensor sees due to the wide FOV of the shadow mask ($F_{dif,FOV} = F_0 \times T_{dif,FOV}$). Equations A3 and A4 form the relationships between $\tau_i$ and $\tau_{cld}$ which we then fit with fourth- or sixth-degree polynomials. The resulting curves give the correction factors for the retrieval outputs, $\tau_{RS,cld}$ and $\tau_{RD}$. The retrieval outputs are the sensor inferred optical depths ($\tau_i$), and
we directly map these to the $\tau_{cld}$ values; these relationships are illustrated in Figure A3.



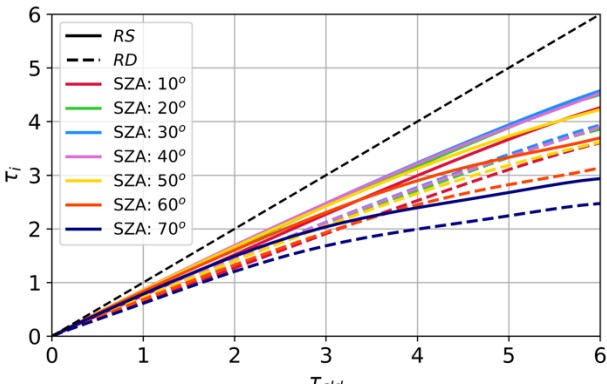

**Figure A3:** $\tau_{cld}$ vs $\tau_i$ relationships for RS and RD (Eqs. A3 and A4) for the 500 nm wavelength, with SZA ranging from $10 - 70^o$.

We derive these $\tau_{cld}$ correction relationship for SZA ranging from $10 - 70^o$ and at wavelengths of 500, 670, and 870 nm. When applying the corrections, the SZA dependence is accounted for by interpolating the derived factors to the SZA at the time of the sample. It is important to note that for the diffuse case, the correction is limited because we ignore the influence a non-zero surface albedo has on $T_{dif,FOV}$. Further, applying the correction to the output of RD when aerosols are present will cause errors, and so the correction is only applied in aerosol free regions. We also ignore the impact ice crystal habit has on this correction because for thin cirrus the overall correction is relatively small (~0-20%), and any error crystal habit induces will therefore have minimal impact on the final retrieved $\tau_{cld}$ value.

*Data availability.* The data can be found in two NASA archives.
ORACLES: https://espo.nasa.gov/ORACLES/archive/browse/oracles/id22/P3.
Camp2ex: https://www-air.larc.nasa.gov/cgi-bin/ArcView/camp2ex.

*Author contributions.* MSN performed the analysis, collected SPN-S and SSFR data, and wrote the manuscript with input from the co-authors. JW developed the SPN-S, helped with interpretation of data, and edited the manuscript. KSS assisted with the development of the methodology, writing and editing of the manuscript. BvD and SAS aided in development of the RD method and edited the manuscript. LDZ, ECC, and MAS collected the CAMP[2]Ex in situ aerosol data, helped with data interpretation, and edited the manuscript. ASK helped with collection of SSFR and SPN-S data. SEL and SB provided 4STAR data and helped with data interpretation and editing. SF collected the ORACLES in situ aerosol. JSR is the PI of the CAMP[2]Ex mission and helped with editing the manuscript.

*Competing interests.* The authors declare that they have no conflict of interest.



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
