# Peer review of "Above-aircraft cirrus cloud and aerosol optical depth from hyperspectral irradiances measured by a total-diffuse radiometer"

_Atmospheric Measurement Techniques, 2021_

## Author Comment (AC1)

We would like to thank the referee for their time and effort in reviewing this manuscript.

Review comments are in blue and the responses by the authors are in black text.

**Referee #1:**

Line 23, remove it within "partition the total optical depth it into"

The change has been made. Sentence now reads:
*Additionally, we use spectral analysis in an attempt to partition the total optical depth into its $\tau_{aer}$ and cirrus cloud optical depth ($\tau_{cld}$) components in the absence of coarse-mode aerosols.*

Line 49-50, a recent study by Yang et al. (2020, doi: 10.1029/2019EA000574) also indicated this point and could be cited as a support.

The reference to Yang et al., 2019, which further supports the notion that clouds interfere with aerosol optical depth retrievals, was added to line 50.

Line 52, "when cirrus is present" is suggested.

The sentence has been updated with "is" replacing "are"
*...though these techniques are limited when cirrus is present (Smirnov et al., 2018).*

Line 61-62, is there any reference to support so high cirrus fraction here? If there is, it is worthy to mention.

The reference to Sassen et al.,. 2008 was cited again, and a reference to Zou et al., (2020; https://doi.org/10.5194/acp-20-9939-2020) was added to support the claim that cirrus have relatively high frequency of occurrence in the tropics.

Line 54-55, it is worthy to mention other types of thin clouds. For example, there are a large fraction of thin clouds in the Arctic with longwave emissivity less than 0.95 as indicated by Garrett et al. (2013, doi: 10.5194/amt-6-1227-2013), who developed a spectral radiation based retrieval algorithm for those thin clouds.

We added text to the end of the paragraph that acknowledges how other thin cloud types cause difficulties when remote sensing aerosols. We also take the opportunity to highlight the approach taken by Garrett and Zhao to study these thin clouds. The Garrett and Zhao paper is also now cited later in the Introduction section in response to a separate comment by the referee.

*Other thin cloud types, such as low-level clouds in the Arctic, can have relatively high rates of occurrence and therefore pose challenges when using remote sensing techniques to study cloud or aerosol optical properties. For the case of thin Arctic clouds, Garrett and Zhao (2013) demonstrated the utility of thermal spectral remote sensing to derive the optical properties when the clouds have an emissivity less than unity.*

Line 77-79, To me, this method is similar to those thin cloud retrieval algorithm that are based on two different band radiation measurements like Garrett et al. (2013) mentioned above.

We now highlight how Garrett and Zhao similarly used the concept a ratio of two spectral measurements to develop a retrieval for thin cloud optical properties. The two methods rely on different physics – in this manuscript the retrievals are based on the study of shortwave radiation, while Garrett and Zhao use transmission and thermal emission, however the point made by the referee is important and we have adjusted the manuscript as follows:

*The concept of using a ratio of two spectral measurements to study thin clouds has been previously developed by Garrett and Zhao (2013) who used a ratio of measured thermal emission to study thermodynamic phase and optical properties of thin Arctic clouds.*

Line 122, how thin the clouds should be to make this equation reliable?

The point we are making by referencing the thin cloud limit (Equation 4) is that the dependence of the diffuse ratio on the asymmetry parameter (g) becomes minimal as clouds become thinner. We feel this point is clear in the manuscript as it is currently written. However, to the referee's point: the measured diffuse ration will have some dependence on g given the presence of any cloud. The question is then at what point does the uncertainty induced from having an unconstrained value of g become insignificant? For the purposes of the Diffuse Method (RD) described in this study, the measurement uncertainty is greater up to optical depts of ~0.5, above this value uncertainty in g becomes a significant source of error. This relationship between the two sources of uncertainties can be seen in Figure 2.

Line 134, I am not sure if dust aerosol has similar absorption characteristics as clouds, please help explain. Thanks.

The use of the term absorption was a mistake. We meant to say that large aerosols and small cloud particles can have similar scattering characteristics -- both can have spectrally flat optical depths at visible wavelengths. It is for this reason that we apply the diffuse method (RD) in the absence of aerosols (because the diffuse signature from large aerosols may look similar to that of clouds). The sentence in line 134 was adjusted as follows:

*Fine-mode aerosols are commonly absorbing, while coarse-mode aerosols, such as sea-salt and dust, have similar scattering characteristics to clouds, both of which limits the application of RD to samples without aerosols (the implications aerosols have on RD are discussed in more detail in Section 5).*

Line 169, have low AE values or have a low AE value?

Thank you for catching this mistake: the word "value" has been changed to the plural form, "values".

Line 255, "One other" should be "Another"? "outputs" or "falls"

This sentence has been corrected in the manuscript.

*Another limitation that is worth mentioning: the output of RD falls below the identity line because of difference in the scattering phase functions used in the simulated data and the DR calculations in the retrieval.*

Line 308, 317 and others, personally, I would like to use "from ... to ..."

We have change how we specify data ranges throughout the manuscript based on the referee's suggestion.

Line 332-335, Why do not the authors correct the diffuse radiation instead of cloud optical depth outputs based on the FOV information?

We correct the optical depths themselves because the bias induced by the FOV is proportional to the optical depth of the cloud. Therefore, given a measurement of the diffuse radiation, finding the magnitude of the bias correction requires that you have information about the cloud optical depth. In our case we use radiative transfer simulations to relate the retrieved optical depth (that's biased by the FOV error) to the un-biased (true) optical depth. Since deriving cloud optical depth is a necessary part of doing the correction, it is simpler to correct the output optical depth rather than correct the diffuse irradiance and then run the retrieval again using this corrected irradiance. This is a similar technique that has been commonly used in past attempts at FOV corrections, for example how FOV bias was handled in Min et al. (2004; doi:10.1029/2003JD003964) or described by Segal-Rosenheimer et al. (2013).

Line 380, "wavelength"

This typo has been corrected.

Figure 4, consistent with other figures, the titles in x- and y- coordinates might start with a capital letter.

The Figure has been updated.

Line 560, "which is shown"?

The word "which" has been added to the sentence. It now reads:

*Like with the ORACLES results, we compare $\tau_{RS,tot}$ to $\tau_{RD}$ for two cases: (1) above the aerosol layer, which is shown in the left panel of Figure 10, and (2) the optical depths for complete spiral profiles (i.e., data from above and within the aerosol layer), which is shown in the right panel of Figure 10.*

Line 613, "related to"

This sentence has been reworked. See the following comment.

Line 613-615, Please rephrase this sentence since it seems with grammar error.

This sentence has been reworked as follows:
*Another source of uncertainty in optical depth retrievals relying on direct/diffuse measurements made from SPN-S is the wide FOV associated with shadow band radiometer systems, which results in an overestimation of the direct transmittance. Biased measurements of transmittances due to the sensor FOV being wider than the solar disk are an issue associated with sun-photometers as well, however the FOV of the SPN-S system is greater than commonly used sun-photometers ($\sim 5 - 10°$ vs $\sim 1°$), making FOV correction necessary for accurate optical depth retrievals (di Sarra et al, 2015; Wood et al, 2017).*

---

## Author Comment (AC2)

We would like to thank the referee for their time and effort in reviewing this manuscript.

Review comments are in blue and the responses by the authors are in black text.

**Referee #2:**

Line 61: "Cirrus presence is especially high in equatorial regions where their frequency of occurrence can be near 50 percent."

A reference would be good here. The cloud occurrence rate will be dependent on the spatial resolution and the sensitivity of a sensor. Does this also include sub-visual cirrus? If yes, then the occurrence rate may be higher?

References to Sassen et al. (2008), and to Zou et al. (2020; https://doi.org/10.5194/acp-20-9939-2020) have been added to text to support the claim about cirrus occurrence in the tropics. We agree with the referee that the presence of sub-visible cirrus can drastically alter the accounting of how often cirrus are present, and therefore different sampling methods can lead to different results. Regardless, cirrus are very common in the tropics, and that is the main point we are trying to make with this sentence. Given that cirrus climatology is not the focus of the manuscript, we have decided to not include a discussion at this level of detail.

Line 79: "This use of the ratio leaves the main sources of uncertainty as the instrument precision and assumptions made in the retrieval itself, resulting in RD being highly sensitive to small variations in optical depth."

I think something is missing here?

This sentence has been reworded:

*By using the ratio of two radiometric quantities as the retrieval input, the measurement uncertainty affecting the retrieval output is strongly dependent on the instrument precision, which is low in relation to the overall radiometric calibration of the system. As a result, the RD method reliably detects thin clouds because it is highly sensitive to small variations in optical depth.*

Line 115, equation (2): How strong an assumption is that? What about full RT-simulations?

We do use full RT simulations when implementing the retrieval method, and these are described in Section 3. The discussion of the two-stream approximation in Section 2 is used in the manuscript to describe to the reader the physics that the retrieval methods rely upon.

Line 116: Perhaps the asymmetry parameter can be defined briefly?

We have adjusted this sentence to include a more detailed description of the asymmetry parameter:

*...where $g$ is the asymmetry parameter of the single atmospheric layer (the mean cosine of the scattering angle, which in the context of the two-stream approximation is a parameter that describes the relative amounts of forward and backwards scattering within a layer)...*

Line 161: "In The" -> "The"

The word "In" has been deleted.

Line 117: "is a term encompassing non-Rayleigh extinction from trace gas molecular scattering"

I'm not sure what "trace gas molecular scattering" means. The Rayleigh scattering by trace gases is negligible. Do you mean absorption by trace gases?

Correct, we are referring to absorption by trace gases. The sentence now reads as follows:

*where $\tau_{Ray,\lambda}$ is the spectral optical depth from Rayleigh scattering, and $\tau_{mol,\lambda}$ is a term encompassing non-Rayleigh extinction from trace gas molecular and water vapor absorption sources.*

Line 192: "For the aerosol-free case within the selected wavelengths region, the δ        profile (dashed blue line) .."

For the dashed blue line the Rayleigh extinction was also removed, I think. Perhaps this should be mentioned explicitly.

We have changed this sentence to explicitly mention that Rayleigh extinction has been removed. The sentence now reads as follows:

*For the aerosol-free case within the selected wavelengths region, the $\tau_\lambda$ profile which accounts for the $\tau_{Ray}$ term (dashed blue line), falls nearly along a flat line of the simulated cloud optical depth value of $\tau_{cld} = 0.20$.*

Lines 217 – 222: What about instrumental straylight, i.e. photons that should not end up on the detector, but get somehow scattered/reflected there? Particularly the diffuse measurements must be affected by that?

The referee brings up an important point about internal reflections due to the shadow mask causing bias in the irradiance measurements. We have adjusted the manuscript to give recognition to how straylight can lead to measurement uncertainty, especially of the diffuse light. However, we do not want to focus too much attention on measurement issues that are specific to this particular version of the SPN-S instrument for a few reasons: (1) The components of the SPN-S were designed and coated with material in an attempt to minimize reflections of light at visible wavelengths. (2) The stated radiometric uncertainties of SPN-S measured irradiances attempt to account for straylight and other design induced errors. Points 1 and 2 are partially addressed in Badosa et al. (2014). (3) SPN-S is a prototype instrument, and it is still undergoing development. Since the design and calibration procedures are currently being advanced, we decided to not focus too much on issues that are related to this specific iteration of the instrument. Rather, in the manuscript, attention was paid to what are traditionally large sources of errors when making airborne irradiance measurements – errors associated with changing sensor attitude, radiometric calibration, shadow mask field-of-view being wider than the solar disk, etc.

The manuscript has been updated with the following text:

*Further, DR is minimally affected by sensor attitude errors because the $F_{dir}$ term is in the denominator. However, there are additional sources of measurement uncertainty related to the SPN-S system that are important to consider, such as internal reflections caused by the shadow mask that can lead to bias in the measured diffuse irradiance. Internal reflections are a problem inherent to total-diffuse radiometer*

*systems reliant on shadow mask (and shadow band) designs, and a detailed discussion of how these, and related, issues have been addressed for the SPN-S are given in Badosa et al. (2014).*

Line 251: "RD outputs have less error .."

What do you mean by "error" here? The difference between retrieved and true values or the retrieval error estimate? This should probably be stated explicitly.

We have rewritten this sentence to make it clearer.

*The results support the analysis of the analytic functions presented in Figure 2: when $\tau_{cld} < \sim 1$ the output of the RD method has less uncertainly in the retrieved optical depth making it the superior method. When $\tau_{cld}$ is greater than unity, there is less uncertainty associated with the retrieved optical depth from the RS method.*

Line 255: "the output of RD fall" -> "the output of RD falls" ?

This sentence has been corrected in the manuscript.

*Another limitation that is worth mentioning: the output of RD falls below the identity line because of difference in the scattering phase functions used in the simulated data and the DR calculations in the retrieval.*

Line 284: "though" -> "through"

Fixed.

Line 293: "Reid et al., (2021)."

Comma after "al." should be removed. Please also check the other citations.

We have removed the extra commas throughout the manuscript.

Line 296: "decent" -> "descent"

Fixed.

Line 322: "DR < 0.9"

At which wavelength?

It is the diffuse ratio is at 500 nm. We have added this information to the text.

Lines 327 – 330: What about contamination of the diffuse measurements by solar photons scattered etc. by the instrument/mask etc.? There must be a contamination to a certain extent?

This comment is related to a previous one made about the text in lines 217-222, and we feel that the issue of straylight contaminating the diffuse measurement has been sufficiently addressed in responding to the

first comment. The paragraph starting at line 327 is focused on sensor FOV induced uncertainty, which of course are related to the shadow mask design of the instrument, but we feel that

Line 380: "wavelenght" -> "wavelength"

Fixed.

Line 383: "angstrom" -> "Angström" (or even better with \AA at the beginning)

We corrected the spelling of Ångström.

Line 411: "DR_mea < 0.95"

At which wavelength?

At 500 nm. The text has been updated.

Caption Fig. 4: "In red, a sample from 1 km which is within the aerosol layer"

The text in the Figure says "below aerosol". Please clarify.

The caption text has been updated to be consistent with Figure legend. The example is from below an aerosol layer.

Line 476: "These optical depths are mostly due to the presence of stratospheric aerosols"

Stratospheric aerosol optical depth in the visible spectral range is on the order of a few times 10^-3 for background conditions, not more, i.e. significantly lower than 0.05.

This statement was worded in a confusing manner. For the retrieval being presented by the manuscript (RS method, which is based on Beer's Law), the AOD above the aerosol layer is 0.05. This non-zero value is mostly due to limitations of the sensitivity of the retrieval method (as is shown in Figures 2 and 3), and not from stratospheric aerosols. (However, a typical stratospheric aerosol AOD value is within the uncertainty of the retrieved AOD, which extends to zero at spiral top.) 4STAR, on the other hand, measures a non-zero AOD above the aerosol layer (typically <0.01) that is due to stratospheric aerosols and trace aerosols in the upper portions of the troposphere less. The paragraph has been reworded for clarity as follows:

*Above the aerosol layer, RD detects a non-zero optical depth of ~0.05, which is mainly a result of the relatively high retrieval uncertainty of the RS method when observing small optical depths (see Figures 2 and 3). This is similar to the measurements made by 4STAR which commonly detects non-zero optical depths at the top of the spiral. However, in the case of 4STAR, these optical depths are mostly due to the presence of stratospheric aerosols (Kremser et al., 2016), though there may also be contribution from small amounts tropospheric aerosol as well.*

Caption Fig. 6: "are sorted into bins with 0.25"

I think "0.25" should be "0.025".

This has been fixed in the manuscript.

Line 552: "However, we expect that since $\tau_{RS,aer}$ is a layer, and not a column optical depth, the forward scattering of light by cirrus which inhibits the SDA method to be less troublesome for our RS method because the irradiance at the top of the layer is directly characterized."

I didn't understand this sentence. Is something missing here? Please adjust or add commas to make the meaning of the statement clearer.

This paragraph has been reworked as follows:

*However, there is reason to expect the RS method will be able to retrieve $\tau_{aer}$ under cirrus conditions to lower uncertainty than has been possible using past methods. For example, the SDA method is developed for use with ground-based measurements and therefore it derives a column optical depth. When retrieving column $\tau_{aer}$ it is difficult to account for the forward scattering of light by cirrus which makes it difficult to accurately constrain $\tau_{aer}$. In the RS method, because the irradiance at the top of the layer is directly characterized, the forward scattering by cirrus is accounted for when deriving $\tau_{aer}$.*

Line 565: "the points generally falling below"

I think "falling" should be "fall", given the first part of the sentence?

Fixed.

Line 574: "data is subsampled" -> "data being subsampled" ?

This sentence was poorly written and it has been rewritten as follows:

*We demonstrate this with an example from the science flight on 20190929 from UTC 03:00 to 05:00. To reduce the computational time required to run the retrieval the 1 Hz SPN-S and SSFR irradiance data is subsampled at 0.1 Hz before being processed.*

Line 577: "preforms" -> "performs"

All incorrect instances of "preforms" have been changed to performs.

[revised manuscript text omitted]